# Score-Optimal Diffusion Schedules

**Christopher Williams**
Department of Statistics
University of Oxford

**Andrew Campbell**
Department of Statistics
University of Oxford

**Arnaud Doucet**
Department of Statistics
University of Oxford

**Saifuddin Syed**
Department of Statistics
University of Oxford

{williams,campbell,doucet,saifuddin.syed}@stats.ox.ac.uk

## Abstract

Denoising diffusion models (DDMs) offer a flexible framework for sampling from high dimensional data distributions. DDMs generate a path of probability distributions interpolating between a reference Gaussian distribution and a data distribution by incrementally injecting noise into the data. To numerically simulate the sampling process, a discretisation schedule from the reference back towards clean data must be chosen. An appropriate discretisation schedule is crucial to obtain high quality samples. However, beyond hand crafted heuristics, a general method for choosing this schedule remains elusive. This paper presents a novel algorithm for adaptively selecting an optimal discretisation schedule with respect to a cost that we derive. Our cost measures the work done by the simulation procedure to transport samples from one point in the diffusion path to the next. Our method does not require hyperparameter tuning and adapts to the dynamics and geometry of the diffusion path. Our algorithm only involves the evaluation of the estimated Stein score, making it scalable to existing pre-trained models at inference time and online during training. We find that our learned schedule recovers performant schedules previously only discovered through manual search and obtains competitive FID scores on image datasets.

## 1 Introduction

Denoising Diffusion models (Sohl-Dickstein et al., 2015; Ho et al., 2020; Song et al., 2021) or DDMs are state-of-the-art generative models. They are formulated through considering a forward noising process that interpolates from the target to a reference Gaussian distribution by gradually introducing noise into an empirical data distribution. Simulating the time reversal of this process then produces samples from the data distribution. Specifically, we evolve data distribution $p_0$ through the *forward diffusion* process on time interval $[0, 1]$, described by

$$\mathrm{d}X_t = f(t)X_t\mathrm{d}t + g(t)\mathrm{d}W_t, \qquad X_0 \sim p_0, \tag{1}$$

with drift $f(t)X_t$, diffusion coefficient $g(t)$ and Brownian noise increment $\mathrm{d}W_t$. The coefficients $f(t)$ and $g(t)$ are chosen such that at time $t = 1$ the distribution of $X_1$ is very close to a reference Gaussian distribution $p_1$ in distribution. A sample starting at $p_0$ and following Equation (1) until time $t$ will be distributed according to $p_t$ which is a mollified version of the data distribution

$$p_t(x_t) = \int_{X_0} p_0(x_0)p_{t|0}(x_t|x_0)\mathrm{d}x_0, \qquad p_{t|0}(x_t|x_0) = \mathcal{N}(x_t; s(t)x_0, \sigma^2(t)I). \tag{2}$$

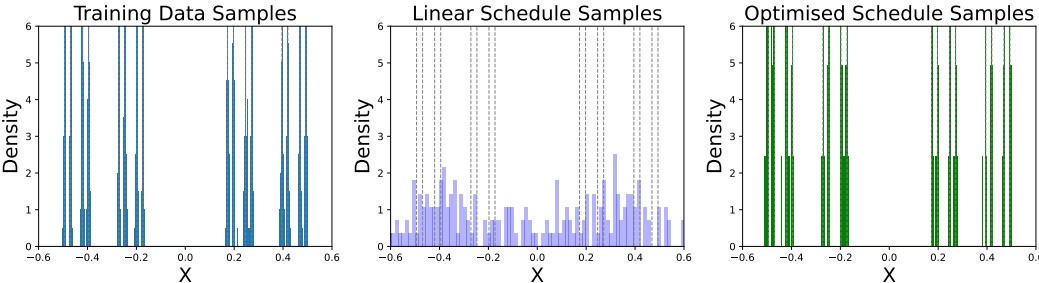

Figure 1: Density estimates of the mollified Cantor distribution (left) using a DDM with schedule $\mathcal{T} = \{t_i\}_{i=0}^{100}$ generated with 100 linearly spaces discretisation times $t_i = i/100$ (middle), compared to the optimised schedule $\mathcal{T}^* = \{t_i^*\}_{i=0}^{50}$ with 50 discretisation times $t_i^*$ generated by Algorithm 1 (right). The eight modes present in our true mollified distribution are shown in grey on each plot.

The parameters $s(t)$ and $\sigma^2(t)$ define the *noising schedule*. They can be found in closed form in terms of $f(t)$ and $g(t)$ (Karras et al., 2022). To obtain samples from $p_0$, we can reverse the dynamics of the forward diffusion in Equation (1) to obtain the *backward diffusion*,

$$\mathrm{d}X_t = \left(f(t)X_t - g(t)^2 \nabla_x \log p_t(X_t)\right)\mathrm{d}t + g(t)\mathrm{d}\widetilde{W}_t, \qquad X_1 \sim p_1. \tag{3}$$

By simulating Equation (3) backwards in time, we evolve reference samples $X_1 \sim p_1$ from $t = 1$ to $t = 0$ to obtain samples that are terminally distributed according to the data distribution $p_0$. To simulate Equation (3) numerically, we must decide upon a discretisation of time, $\mathcal{T} = \{t_i\}_{i=0}^{T}$ with $t_T = 1$, $t_0 = 0$, which we refer to as the *discretisation schedule*. For a given noising schedule, it is important to select an appropriate discretisation schedule such that (3) can be simulated accurately, i.e. $p_{t_i}$ and $p_{t_{i-1}}$ should not differ significantly. In this paper, we derive a methodology to compute an optimal discretisation schedule.

Prior work has often joined together the choice of noising schedule and discretisation schedule. A uniform splitting of time would be chosen, $t_i = i/T$, with the noising schedule dictating the change between $p_{t_i}$ and $p_{t_{i-1}}$. Two prominent examples have the form $s(t) = \sqrt{\bar{\alpha}(t)}$, $\sigma^2(t) = 1 - \bar{\alpha}(t)$ with the *linear schedule* introduced by Ho et al. (2020) having $\bar{\alpha}(t) = 1 - \exp\left(-\int_0^t \beta(s)\mathrm{d}s\right)$ with linear $\beta(t) = \beta_{\min} + t(\beta_{\max} - \beta_{\min})$. Alternatively, Nichol and Dhariwal (2021) introduce the *cosine schedule* with $\bar{\alpha}(t) = f(t)/f(0)$, $f(t) = \cos\left(\frac{t+\epsilon}{1+\epsilon}\frac{\pi}{2}\right)^2$ for $\epsilon = 0.008$. Karras et al. (2022) refines this approach by splitting the choice of noising schedule from that of the discretisation schedule, however, picking the discretisation schedule is still a matter of hyperparameter tuning.

A good discretisation schedule can drastically impact the efficiency of the training and inference of the generative model, but unfortunately can be difficult to select for complex target distributions. For example, for a distribution supported on the Cantor set (Figure 1, left), the default linear schedule fails entirely to capture the modes of the data distribution (Figure 1, middle). However, our optimised schedule learned using Algorithm 1 recovers these modes (Figure 1, right). Without such an automatic algorithm, finding performant discretisation schedules often reduces to an expensive and laborious hyperparameter sweep.

We devise a method for selecting a discretisation schedule that yields high-quality samples from $p_0$. Our key contribution is defining an appropriate notion of cost incurred when simulating from one step in the diffusion path to the next. We then choose our discretisation schedule to minimise the total cost incurred when simulating the entire path from $p_1$ to $p_0$. Our cost purely depends on the distributional shifts along the diffusion path and assumes perfect score estimation, hence, we refer to our schedules as *score-optimal diffusion schedules*. The resulting algorithm is cheap to compute, easy to implement and requires no hyperparameter search. Our algorithm can be applied to find discretisation schedules for sampling pre-trained models as well as performed online during DDM training. We demonstrate our proposed method on highly complex 1D distributions and show our method scales to high dimensional image data where it recovers previously known performant discretisation schedules discovered only through manual hyperparameter search. To the best of our knowledge, this is the first online data-dependent adaptive discretisation schedule tuning algorithm.

## 2 The Cost of Traversing the Diffusion Path

To derive our optimal discretisation schedule, we first need to derive a notion of cost of traversing from our reference distribution $p_1$ to the data distribution $p_0$ through each intermediate distribution $p_t$, referred to as the diffusion path. We will then later find the discretisation schedule that minimises the total cost of traversing this path. Our notion of cost is based on the idea that while integrating Equation (3) from time $t$ to $t'$ will always take you from $p_t$ to $p_{t'}$, the simulation step will need to do more work to make sure the samples are distributed according to $p_{t'}$ if $p_t$ and $p_{t'}$ are very different distributions rather than if they are close. In the following, we will make this intuition precise.

### 2.1 Predictor/Corrector Decomposition of the Diffusion Update

To begin, let $\mathbb{X} = \mathbb{R}^d$ and $\mathbb{P}(\mathbb{X})$ be the space of Lebesgue probability measures on $\mathbb{X}$ with smooth densities. For $t \in [0, 1]$, define the diffusion path $p_t \in \mathbb{P}(\mathbb{X})$ as the law of $X_t$ satisfying the forward diffusion Equation (1) initialised at the data distribution $X_0 \sim p_0$, or equivalently the law of the $X_t$ satisfying the backward diffusion Equation (3) initialised at the reference distribution $X_1 \sim p_1$.

Given a sample $X_t \sim p_t$, a sample $X_{t'} \sim p_{t'}$ can be generated by integrating the backward diffusion Equation (3). In Song et al. (2021), it was shown that we can further decompose the backward diffusion Equation (3) at time $t$ into a deterministic flow governed by the probability flow ODE, and the stochastic flow driven by Langevin dynamics targeting $p_t$,

$$\mathrm{d}X_t = \underbrace{\left(f(t)X_t - \tfrac{1}{2}g(t)^2 \nabla \log p_t(X_t)\right)\mathrm{d}t}_{\text{Probability Flow Prediction ODE}} + \underbrace{-\tfrac{1}{2}g(t)^2 \nabla \log p_t(X_t)\mathrm{d}t + g(t)\mathrm{d}\widetilde{W}_t}_{\text{Langevin Correction SDE}}. \quad (4)$$

This decomposition into a deterministic flow and a correction will help us derive our cost in Section 2.2 by analysing the work done by the correction to keep the samples on the diffusion path. Here, we will first expand upon this decomposition by defining a hypothetical two-step sampling procedure that could be used to sample the DDM. It consists of: (1) a *predictor step* that generates a deterministic prediction of $X_{t'}$ and (2) *a corrector step* that uses Langevin dynamics targeting $p_{t'}$ to correct any accrued error. Note we are not advocating for the implementation of such a procedure, only that by imagining simulating with this hypothetical predictor-corrector algorithm, it will be helpful for our theoretical derivation of a cost intrinsically linked to the sampling of DDMs. The two stages of the predictor-corrector algorithm are rigorously defined as follows.

**Definition 2.1.** *A predictor for $t, t' \in [0, 1]$ is a smooth bijective mapping $F_{t,t'} : \mathbb{X} \mapsto \mathbb{X}$, such that* $\det \nabla F_{t,t'} \neq 0$*, and the predicted distribution is the pushforward of $p_t$ by $F_{t,t'}$ denoted $F_{t,t'}^\sharp p_t$.*

**Definition 2.2.** *A corrector for $t \in [0, 1], \tau \in [0, \infty)$ is a one-parameter family Markov transition kernel $L_{t,\tau} : \mathbb{X} \times \mathbb{P}(\mathbb{X}) \mapsto [0, 1]$ such that $Z_\tau \sim L_{t,\tau}(z, \mathrm{d}z_\tau)$ is the law of Langevin dynamics stationary distribution $p_t$ at time $\tau$, initialised at $z \in \mathbb{X}$, and running at speed $v(t) > 0$,*

$$Z_0 = z, \quad \mathrm{d}Z_\tau = v(t)\nabla \log p_t(Z_\tau)\mathrm{d}\tau + \sqrt{2v(t)}\mathrm{d}W_\tau. \quad (5)$$

The corrector map $L_{t,\tau}$ is specified through an integration time $\tau$ and time-dependent speed $v(t)$. We assume $\tau$ is fixed and we will describe the appropriate choice of $v(t)$ in Section 3.3. The predictor-corrector algorithm, given $X_t \sim p_t$, first applies the predictor $Z_0 = F_{t,t'}(X_t)$ and then uses the corrector to drive the predicted samples towards $p_{t'}$,

$$\text{Predictor:} \quad Z_0 = F_{t,t'}(X_t) \qquad \text{Corrector:} \quad X_{t',\tau} \sim L_{t',\tau}(Z_0, \mathrm{d}x_\tau). \quad (6)$$

In general, $Z_0$ will not be a sample from $p_{t'}$ exactly because $F_{t,t'}$ may not be a perfect transport from $p_t$ to $p_{t'}$. In Section 2.2, assessing the work done by $L_{t',\tau}$ to drive the $Z_0$ towards $p_{t'}$ will be key in deriving our cost. Our cost will then depend upon the specific choice for $F_{t,t'}$. Two natural choices for $F_{t,t'}$ are apparent. Setting $F_{t,t'}$ to the identity means our hypothetical sampling algorithm reduces to the annealed Langevin algorithm for DDMs introduced by Song and Ermon (2019). The second natural choice is to set $F_{t,t'}$ to the integrator of the probability flow ODE (Song et al., 2021).

**Example 2.1** (Annealed Langevin). The predictor step is trivial when the predictor map is the identity $F_{t,t'}(x) = x$. In such a case, the predicted state reduces to the initial state, $F_{t,t'}(X_t) = X_t \sim p_t$. The work done by the corrector step will then be related to the full discrepancy between $p_t$ and $p_{t'}$ because the predictor provides no help in transporting the sample.

**Example 2.2** (Probability Flow ODE). The predictor step is optimal when $F_{t,t'}$ is a transport from $p_t$ to $p_{t'}$. In such a case, the predicted state produces a sample from the target distribution, $F_{t,t'}(X_t) \sim p_{t'}$, and so the corrector step would have to perform no work. An optimal predictor map $F_{t,t'}$ can be obtained by integrating the probability flow ODE from time $t, t'$,

$$\frac{\mathrm{d}x_t}{\mathrm{d}t} = f(t)x_t - \frac{1}{2}g(t)^2 \nabla \log p_t(x_t). \tag{7}$$

Practical algorithms numerically integrate Equation (7), e.g. an Euler step with $\Delta t = t' - t$,

$$F_{t,t'}(x) = x + \left(f(t)x - \tfrac{1}{2}g(t)^2 \nabla \log p_t(x)\right)\Delta t. \tag{8}$$

In such a case, the work done by the corrector depends on the error in the probability flow integrator.

## 2.2 The Incremental Cost of Correction

We now focus on deriving a cost related to the work done by the corrector step in the predictor-corrector algorithm. Later, in Section 3, we will find the discretisation schedule that minimises the total cost. To derive the cost, we will analyse the movement of $Z_0$ under the corrector step's dynamics $L_{t',\tau}(Z_0, \mathrm{d}x_\tau)$. This requires some care because even if $Z_0$ is already at stationarity, i.e. perfectly distributed according to $p_{t'}$, applying the Langevin correction step will still result in movement of $Z_0$ due to the stochasticity of the update. However, the computed *work* done by the correction step in this case should be 0. To correctly assign the *work* done, we will compare two processes. The first is the trajectory of Langevin dynamics, $Z_\tau$, defined by the corrector $L_{t',\tau}$ initialised at $Z_0 = F_{t,t'}(X_t)$ targeting $p_{t'}$. The second is a virtual coupled Langevin dynamics $\tilde{Z}_\tau$ initialised at $F_{t,t'}(X_t)$, driven by the same noise and speed but targeting the stationary distribution of the predictor $F^\sharp_{t,t'}p_t$,

$$Z_0 = F_{t,t'}(X_t), \quad \mathrm{d}Z_\tau = v(t')\nabla \log p_{t'}(Z_\tau)\mathrm{d}\tau + \sqrt{2v(t')}\mathrm{d}W_\tau, \tag{9}$$

$$\tilde{Z}_0 = F_{t,t'}(X_t), \quad \mathrm{d}\tilde{Z}_\tau = v(t')\nabla \log F^\sharp_{t,t'}p_t(\tilde{Z}_\tau)\mathrm{d}\tau + \sqrt{2v(t')}\mathrm{d}W_\tau. \tag{10}$$

Notably, $Z_\tau \overset{d}{=} X_{t',\tau}$ and $\tilde{Z}_\tau \overset{d}{=} F_{t,t'}(X_t)$ share the same law as the corrected sample and predicted sample respectively. Since $Z_\tau$ and $\tilde{Z}_\tau$ are coupled to have the same noise, the difference in their trajectory, $Z_\tau - \tilde{Z}_\tau$, isolates the change in corrector dynamics due to discrepancy between $F^\sharp_{t,t'}p_t$ and $p_{t'}$. If $F^\sharp_{t,t'}p_t$ is very different to $p_{t'}$, then $Z_\tau - \tilde{Z}_\tau$ will be large, signifying the corrector is needing to do lots of work to push the distribution of $Z$ towards the target $p_{t'}$. Conversely, if $F^\sharp_{t,t'}p_t = p_{t'}$, then $Z_\tau - \tilde{Z}_\tau = 0$ and no work is done. For small $\tau$, $(Z_\tau - Z_0)/\tau$ is the initial velocity of $Z$ under the $p_{t'}$ corrector dynamics, and similarly for $(\tilde{Z}_\tau - Z_0)/\tau$. We can then define the *incremental cost* $\mathcal{L}(t, t')$ by taking limits as $\tau \to 0^+$, measuring the expected $L^2$ norm $\|\cdot\|$ of the difference,

$$\mathcal{L}(t, t') = \lim_{\tau \to 0^+} \tau^{-2} \mathbb{E}\left[\left\|(Z_\tau - Z_0) - (\tilde{Z}_\tau - Z_0)\right\|^2\right] = \lim_{\tau \to 0^+} \tau^{-2}\mathbb{E}\left[\left\|Z_\tau - \tilde{Z}_\tau\right\|^2\right]. \tag{11}$$

We can approximate $Z_\tau - \tilde{Z}_\tau$ using an Euler step, noting that the coupled noise terms cancel,

$$Z_\tau - \tilde{Z}_\tau = \tau v(t')(\nabla \log p_{t'}(Z_{t,t'}) - \nabla \log F^\sharp_{t,t'}p_t(Z_{t,t'})) + o(\tau). \tag{12}$$

By substituting Equation (12) in Equation (11), we have

$$\mathcal{L}(t, t') = v(t')^2 \mathbb{E}\left[\left\|\nabla \log p_{t'}(Z_0) - \nabla \log F^\sharp_{t,t'}p_t(Z_0)\right\|^2\right] = v(t')^2 D(p_{t'}\|F^\sharp_{t,t'}p_t), \tag{13}$$

where $D(p\|q) = \mathbb{E}_{X \sim q}[\|\nabla \log p(X) - \nabla \log q(X)\|^2]$ is a statistical divergence on $p, q \in \mathbb{P}(\mathbb{X})$, measuring the $L^2$ distance between the scores of $q$ and $p$ with respect $q$. $D(p\|q)$ is referred to as the Stein divergence or the Fisher divergence; see e.g. (Johnson, 2004). For a given choice of $v(t)$ and $F_{t,t'}$ we now have a cost measuring the change from $p_t$ to $p_{t'}$. This cost is intrinsically linked with the effort performed by a DDM sampling algorithm because it is derived through considering the work done by a hypothetical predictor-corrector style update. We note, however, that this general cost can be used to obtain discretisation schedules for use in any style of DDM sampler.

## 2.3 Corrector and Predictor Optimised Cost

By inverting $Z_0 = F_{t,t'}(X_t)$, we can express Equation (13) in terms of an expectation with respect to the reference sample $X_t \sim p_t$, and the score of $G_{t,t'} : \mathbb{X} \mapsto \mathbb{R}_+$, the incremental weight function associated with the transport $F_{t,t'}$ from the Sequential Monte Carlo literature (Arbel et al., 2021),

$$\mathcal{L}(t,t') = v(t')^2 \mathbb{E}\left[\|\nabla \log G_{t,t'}(X_t)\|^2\right], \quad G_{t,t'}(x) = \frac{p_{t'}(F_{t,t'}(x))}{p_t(x)}|\det \nabla F_{t,t'}(x)|. \quad (14)$$

In most cases, it is infeasible to efficiently compute the Jacobian correction in Equation (14). When $F_{t,t'}(x) = x$ is the identity map corresponding to the corrector optimised update from Example 2.1 Equation (14) reduces a rescaled Stein discrepancy between $p_t$ and $p_{t'}$, and $G_{t,t}(x) = p_{t'}(x)/p_t(x)$ reduces to the likelihood-ratio between $p_{t'}$ and $p_t$. We will refer to this case as the *corrector-optimised cost* denoted $\mathcal{L}_c(t,t')$, to distinguished it from the *predictor-optimised cost* $\mathcal{L}_p(t,t')$ derived above, where when relevant, we will use subscripts $c$ and $p$ to distinguish between the two:

$$\mathcal{L}_c(t,t') = v(t')^2 D(p_{t'}\|p_t), \qquad \mathcal{L}_p(t,t') = v(t')^2 D(p_{t'}\|F_{t,t'}^\sharp p_t). \quad (15)$$

The corrector-optimised cost $\mathcal{L}_c(t,t')$ provides meaningful information during the update from reference $p_t$ to the target $p_{t'}$. It is worth computing even when the predictor-optimised cost $\mathcal{L}_p(t,t')$ is accessible. $\mathcal{L}_c(t,t')$ measures the change between the reference and target distribution independent of the predictor, whereas $\mathcal{L}_p(t,t')$ measures the residual error between the predictor and target. Notably, $\mathcal{L}_c(t,t')$ encodes information about the incremental geometry of the diffusion path, whereas $\mathcal{L}_p(t,t')$ quantifies information about the incremental efficiency of the predictor. Generally, one does not dominate the other, but if the predictor is well-tuned and the predictor flows samples $X_t \sim p_t$ towards $p_{t'}$, we would expect $\mathcal{L}_p(t,t') \leq \mathcal{L}_c(t,t')$.

For deriving our optimal discretisation schedule, we require a notion of how $\mathcal{L}(t,t')$ increases with small increases in $t'$ i.e. knowing local changes in incremental cost. In Section 3, we use this *local cost* to assign *distances* to schedules through time, enabling us to find the best schedule. We derive the desired local cost in Theorem 2.1, see Appendix A for a PDE and geometric interpretation.

**Theorem 2.1.** *Suppose $p_t(x), F_{t,t'}(x), v(t)$ and $G_{t,t'}(x)$ are three-times continuously differentiable in $t, t', x$ and let $\dot{F}_t(x) = \frac{\partial}{\partial t'} F_{t,t'}(x)\big|_{t'=t}$ and $\dot{G}_t(x) = \frac{\partial}{\partial t'} G_{t,t'}(x)\big|_{t'=t}$. Suppose the following hold: (1) for all $x \in \mathbb{X}, t \in [0,1]$, $F_{t,t}(x) = x$ and (2) there exists $V : \mathbb{X} \mapsto \mathbb{R}$ such that for all $x \in \mathbb{X}$ and $t \in [0,1]$, $\|\nabla \dot{G}_t(x)\|^2 \leq V(x)$ and $\sup_{t \in [0,1]} \mathbb{E}_{X_t \sim p_t}[V(X_t)] < \infty$. Then for all $t \in [0,1]$, we have $\mathcal{L}(t,t') = \delta(t)\Delta t^2 + O(\Delta t^3)$, where*

$$\delta(t) = v(t)^2 \mathbb{E}_{X_t \sim p_t}\left[\left\|\nabla \dot{G}_t(X_t)\right\|^2\right], \quad \dot{G}_t = \frac{\partial}{\partial t} \log p_t + \nabla \log p_t \cdot \dot{F}_t + \mathrm{Tr}\nabla \dot{F}_t. \quad (16)$$

Theorem 2.1 shows that, under regularity assumptions, then the incremental cost is $\mathcal{L}(t,t') \approx \delta(t)\Delta t^2$ is locally quadratic and controlled by the local cost $\delta(t)$. The $\delta(t)$ measures the sensitivity of the incremental cost $\mathcal{L}(t,t')$ to moving samples along the diffusion path to $t' \approx t$. Notably, $\delta(t) = 0$ if and only if the predictor satisfies the continuity equation, $\frac{\partial}{\partial t} p_t + \nabla \cdot (p_t \dot{F}_t) = 0$.

## 3 Score-Optimal Schedules

Given a discretisation schedule $\mathcal{T} = (t_i)_{i=0}^T$ satisfying $0 = t_0 < \cdots < t_T = 1$, our hypothetical predictor-corrector algorithm recursively uses the predictor and corrector maps to generate a sequence $(X_i)_{i=0}^T$ starting at $X_T \sim p_1$ such that the terminal state $X_0$ approximates samples from $p_0$,

$$X_i \sim L_{t_i,\tau}(F_{t_{i+1},t_i}(X_{i+1}), \mathrm{d}x_i). \quad (17)$$

We want to identify a discretisation schedule that maximises the efficiency of this iterative procedure. This is not generally possible due to the potential complex interactions that arise from the accrued errors. To simplify our analysis, we make the following assumption.

**Assumption 3.1.** *For all $t, t'$, if $X_t \sim p_t$ and $X_{t',\tau} = L_{t',\tau}(F_{t,t'}(X_t), \mathrm{d}x_\tau)$, then $X_{t',\tau} \sim p_{t'}$.*

Assumption 3.1 is reasonable if, in our hypothetical corrector steps, $\tau$ is set sufficiently large such that the Langevin correction converges to stationarity. We find in our experiments that even if the

schedules derived under Assumption 3.1 are used in sampling algorithms for which Assumption 3.1 does not hold, we still obtain high quality samples. Equipped with Assumption 3.1, we can measure the efficiency of the path update through total accumulated cost $\mathcal{L} = \sum_{i=1}^{T} \mathcal{L}(t_{i+1}, t_i)$, which we will use as our objective to optimise $\mathcal{T}$. In this section, we will identify the optimal schedule $\mathcal{T}^*$ minimising the cost $\mathcal{L}$ by considering an infinitely dense limit. We will then provide a tuning procedure amenable to online schedule optimisation during training. Finally, we will discuss a suitable choice for $v(t)$, the velocity of our hypothetical corrector steps, as well as related work.

### 3.1 Diffusion Schedule Path Length and Energy

Let $\varphi : [0,1] \mapsto [0,1]$ be a strictly increasing, differentiable function such that $\varphi(0) = 0$ and $\varphi(1) = 1$. We will say $\mathcal{T}$ is generated by $\varphi$ if $t_i = \varphi(i/T)$ for all $i = 0, \ldots, T$. The schedule generator $\varphi$ dictates how fast our samples move through their diffusion path. Since every schedule $\mathcal{T}$ of size $T$ is generated by some $\varphi$, optimising $\mathcal{T}$ is equivalent to finding a generator $\varphi$ minimising $\mathcal{L}(\varphi, T)$, the total cost accumulated by the schedule of size $T$ generated by $\varphi$. By Jensen's inequality, we have $\mathcal{L}(\varphi, T) \geq \Lambda(\varphi, T)^2/T$, where for $t_i = \varphi(i/T)$,

$$\mathcal{L}(\varphi, T) := \sum_{i=1}^{T} \mathcal{L}(t_{i+1}, t_i), \quad \Lambda(\varphi, T) = \sum_{i=1}^{T} \sqrt{\mathcal{L}(t_{i+1}, t_i)}. \tag{18}$$

As we later prove in Theorem 3.1, in the dense schedule limit as $T \to \infty$, the cost $\mathcal{L}(\varphi, T)$ and its lower bound $\Lambda(\varphi, T)$ are controlled by the *energy* $E(\varphi)$ and *length* $\Lambda$ respectively where,

$$E(\varphi) = \int_0^1 \delta(\varphi(s))\dot{\varphi}(s)^2 \mathrm{d}s, \quad \Lambda = \int_0^1 \sqrt{\delta(t)} \mathrm{d}t. \tag{19}$$

The intuition for why $E(\varphi)$ is an energy, and $\Lambda$ a length can be gained by first conceptualising the diffusion time $t$ as a spatial variable rescaled by the metric $\delta(t)$ defined by our cost $\mathcal{L}$. We have $\varphi$ and $\dot{\varphi}$ are position and velocity, respectively. Integrating the speed $\int_0^1 \sqrt{\delta(\varphi(s))}\dot{\varphi}(s)\mathrm{d}s = \int_0^1 \sqrt{\delta(t)}\mathrm{d}t$ along a curve $\varphi(s)$ obtains the "length" $\Lambda$, whilst integrating a speed squared, $\int_0^1 \delta(\varphi(s))\dot{\varphi}(s)^2 \mathrm{d}s$ obtains a "kinetic energy" $E(\varphi)$. Note that the length is an invariant of the schedule, whereas the kinetic energy is not. The length $\Lambda$ measures the intrinsic difficulty of traversing the diffusion path according to the cost independent of $\varphi$, whereas $E(\varphi)$ measures the efficiency of how the path was traversed using $\varphi$. This geometric intuition hints at the solution to the optimal scheduling problem. The optimal $\varphi$ should travel on a geodesic path from $p_1$ to $p_0$, at a constant speed with respect to metric $\delta$. For this optimal $\varphi$, we then have the kinetic energy being equal to the square of length between $p_1$ and $p_0$. Theorem 3.1 makes the previous discussion precise.

**Theorem 3.1.** *Suppose the assumptions of Theorem 2.1 hold. For all schedule generators $\varphi$,*

$$\lim_{T \to \infty} T\mathcal{L}(\varphi, T) = E(\varphi), \quad \lim_{T \to \infty} \Lambda(\varphi, T) = \Lambda. \tag{20}$$

*Moreover, $E(\varphi) \geq \Lambda^2$, with equality if and only if $\varphi^*$ satisfies,*

$$\varphi^*(s) = \Lambda^{-1}(\Lambda s), \quad \Lambda(t) = \int_0^t \sqrt{\delta(u)} \mathrm{d}u. \tag{21}$$

Notably independent of the choice of $\varphi$, as $T \to \infty$, the cost $\mathcal{L}(\varphi, T) \sim E(\varphi)/T$. This implies that the cost decays to zero at a linear rate, proportional to $E(\varphi)$ and $\mathcal{L}(\varphi, T) \gtrsim \Lambda^2/T$ independent of $\varphi$. Equation (21) provides an explicit formula for the optimal schedule generator that minimises the dense limit of the total cost and obtains the lower bound $E(\varphi^*) = \Lambda^2$. The intuition for the formula $\varphi^*(s) = \Lambda^{-1}(\Lambda s)$ is that this implies $\Lambda(\varphi^*(s)) = \Lambda s$ meaning say 10% of the way through the optimal schedule, we should have traversed 10% of the way along the distance between $p_1$ and $p_0$ i.e. $0.1 \times \Lambda$. This relation holds for constant speed straight lines, meaning $\varphi^*$ is the optimal schedule. For a finite $T$, Theorem 2.1 implies the optimal schedule $\mathcal{T}^* = \{t_i^*\}_{i=0}^{T}$ generated by $\varphi^*$ ensures the incremental cost is constant $\mathcal{L}(t_{i+1}^*, t_i^*) \approx \Lambda^2/T^2$ for all $i = 0, \ldots, T-1$.

Our geometric intuition in the language of differential geometry is that the diffusion path $\mathcal{M} = \{p_t\}_{t \in [0,1]}$ is Riemannian manifold with metric $\delta$ endowed by the incremental cost $\mathcal{L}(t, t')$. The schedule generator defines a curve $s \mapsto p_{\varphi(s)} \in \mathcal{M}$ reparametrising the diffusion path between $p_0$ and $p_1$. Theorem 3.1 shows that $\varphi^*$ is the geodesic of length $\Lambda$ in $\mathcal{M}$ between $p_1$ and $p_0$ that traverses the diffusion path at a constant speed $\sqrt{\delta(\varphi^*(s))}\dot{\varphi}^*(s) = \Lambda$ with respect to $\delta$ and minimises the cost.

## 3.2 Estimation of Score-Optimal Schedules

Given a schedule $\mathcal{T} = \{t_i\}_{i=0}^{T}$ and estimates of the incremental cost $\mathcal{L}(t_{i+1}, t_i)$, Algorithm 1 adapts Algorithm 3 from Syed et al. (2021) to estimate the optimal schedule $\mathcal{T}^* = \{t_i^*\}_{i=0}^{T}$ generated by $\varphi^*$. We can use Algorithm 1 to refine the schedule for a pre-trained DDM or learn the schedule jointly with the score function. For this joint procedure, we detail in Appendix B.1 how function evaluations can be reused to estimate the cost to minimise computational overhead. For $\mathcal{L}_c(t, t')$ we need only evaluate $\nabla \log p_t(X_t)$ and $\nabla \log p_{t'}(X_t)$ both available through our model's score estimate. Computing $\mathcal{L}_p(t, t')$ is more challenging since there are Hessian terms that arise in Equation (14). Under the assumption that the step size $\Delta t > 0$ is sufficiently small, we can approximate $\nabla \log | \det \nabla F_{t,t'}(X_t)|$ through Proposition B.1. This approximation only requires us to compute the gradient trace of the Jacobian of our predicted score. With computational cost proportionate to the computational effort for computing the first derivative. Using a Hutchinson trace (Hutchinson, 1989) like estimator in Proposition B.1, we compute this quantity memory-efficiently in high dimensions, requiring only standard auto-differentiation back-propagation.

## 3.3 Choice of Velocity Scaling

Recall that our cost is derived by considering a Langevin dynamics step with velocity $v(t)$. This velocity should be selected so that Langevin dynamics explores the same proportion of our distribution at varying times throughout our diffusion path. Thus, $v(t)$ should be on the same scale as the spread of the target, $p_t$. Commonly used noising schedules have $s(t) \leq 1$, and our data distribution is normalised so the scale of $p_t$ is on the order of $\sigma(t)$. We therefore set $v(t) = \sigma(t)$. This results in a $\sigma(t')$-weighted divergence for our incremental cost $\mathcal{L}(t, t') = \sigma(t')^2 D(p_{t'}||p_t)$. This can be compared to the weighted denoising score matching loss used to train DDMs (Song et al., 2021), which is also a squared norm of score differences: $\lambda(t)\mathbb{E}_{X_0,X_t}\left[\|s_\theta(X_t, t) - \nabla \log p_{t|0}(X_t|X_0)\|^2\right]$ for some weighting function $\lambda(t)$ chosen to equalise the magnitude of the cost over the path. In Song et al. (2021), $\lambda(t) \propto 1/\mathbb{E}[\|\nabla \log p_{t|0}(X_t|X_0)\|^2]$ was chosen, which, as we show in Appendix B.3, is $\lambda(t) \propto \sigma^2(t)$. This choice of velocity scaling provides an alternative perspective on this commonly used weighting of squared norms of score differences.

---

**Algorithm 1** `UpdateSchedule`

---

**Require:** Schedule $\mathcal{T} = \{t_i\}_{i=0}^{T}$, incremental costs $\{\mathcal{L}(t_{i+1}, t_i)\}_{i=0}^{T-1}$
1: $\hat{\Lambda}(t_i) = \sum_{j=0}^{i-1} \sqrt{\mathcal{L}(t_{j+1}, t_j)}, \quad i = 0, \ldots T$           ▷ Equation (21);
2: $\hat{\Lambda} = \hat{\Lambda}(t_T)$                  ▷ $\Lambda$ in Equation (19)
3: $\hat{\Lambda}^{-1}(\cdot) = \texttt{Interpolate}(\{(\hat{\Lambda}(t_0), t_0), \ldots, (\hat{\Lambda}(t_T), t_T)\});$    ▷ E.g. Fritsch and Carlson (1980)
4: $t_i^* = \hat{\Lambda}^{-1}(\hat{\Lambda}\frac{i}{T}), \quad i = 0, \ldots, T$             ▷ Equation (21)
5: **Return:** $\mathcal{T}^* = \{t_i^*\}_{i=0}^{T}$

---

## 3.4 Related Work

Previous works have devised algorithms and heuristics for designing noising and discretisation schedules. The DDM training objective is invariant to the noising schedule shape, as demonstrated by Kingma et al. (2021), necessitating auxiliary costs and objectives for schedule design. Uniform steps in the signal-to-noise ratio, $\log(s(t_i)/\sigma(t_i))$, are used by Lu et al. (2022), but this ignores the target distribution's geometry. Watson et al. (2021) optimise the schedule by differentiating through sampling to maximise quality, but GPU memory constraints necessitate gradient rematerialisation. We avoid this with a simulation-free cost. Closely related to our work is Sabour et al. (2024), who minimise a pathwise KL-divergence between discretised and continuous processes. They require multi-stage optimisation with early stopping to prevent over-optimisation of their objective which would otherwise result in worse schedules. Amongst the wider literature, various strategies for discretisation schedule tuning have been proposed. Das et al. (2023) derive an equally spaced schedule using the Fisher metric but assume Gaussian data. Santos et al. (2023) assign time points proportional to the Fisher information of $p_{t|0}(x_t \mid x_0)$, ignoring the true target distribution. Xue et al. (2024) derive a schedule to control ODE simulation error, but their cost depends only on the ODE solver, and not on the data distribution.

## 4 Computational Experiments

### 4.1 Sampling the Mollified Cantor Distribution

The Cantor distribution (Cantor, 1884) lacks a Lebesgue density, with its cumulative distribution function represented by the Devil's staircase and its support being the Cantor set, forming a challenging 1-D test example. When mollified with Gaussian noise, it becomes absolutely continuous and possesses a Stein score. We mollify by running a diffusion with the linear schedule for time $t = 10^{-5}$. With this mollification, our data density has eight pronounced peaks. We train a one-dimensional DDM for 150,000 iterations using both a fixed linear schedule and our optimisation algorithm Algorithm 2 initialised at the linear schedule. We find that the non-data-specific default schedule fails to capture these modes, whilst our adaptive method faithfully reproduces the data distribution. In Figure 7 we show the complexity of the learned score which displays a self-similar fractal structure.

### 4.2 Adaptive Schedule Learning for Bimodal Example

We train a DDM on a simple bimodal Gaussian distribution. When the variance of the target bimodal Gaussian is low, it becomes difficult to adequately sample from the target distribution. In our instance, the standard Gaussian reference from the diffusion is given, and the target is the density $p_0(x) = \frac{1}{2}p_{\text{left}}(x) + \frac{1}{2}p_{\text{right}}(x)$, where $p_{\text{left}}$ and $p_{\text{right}}$ are normal distributions with means $-6$ and $6$, respectively, and a common variance $\sigma^2 = 0.1^2$.

We learn two diffusion models, one using the linear schedule, and the other using a schedule that is learned online during training. We compute the likelihood of the samples generated from either model during training, which is possible in this example because the true probability density is known. It can be seen in Figure 2 that when the schedule is learned during training, the likelihood evaluation increases and the true score error decreases, in contrast to the linear schedule that remains constant, or worsens, in this regard during training.

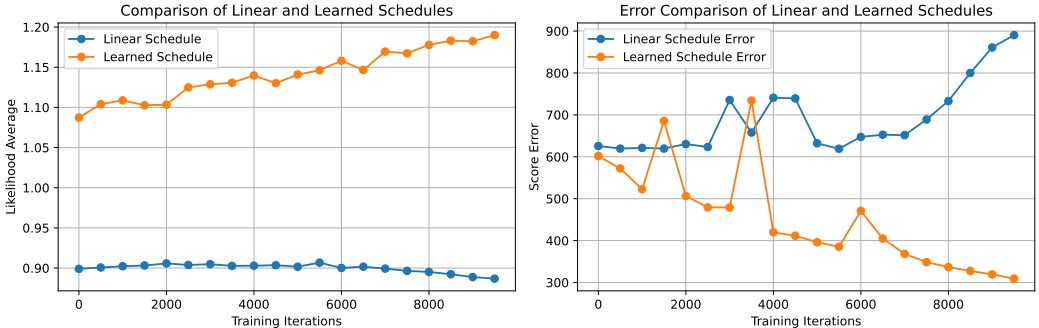

Figure 2: Comparison of Linear and Learned Schedules over Training Iterations for the bimodal example. Each point corresponds to 500 training iterations.

### 4.3 Scalable Schedule Learning Diffusion

Here we demonstrate that jointly learning the schedule and score using our online training methodology (Algorithm 2) scales to high-dimensional data and converges to a stable solution. We train DDMs on CIFAR-10 and MNIST initialised at the cosine schedule using the codebase from Nichol and Dhariwal (2021). In Figure 3 (left), we show the incremental costs $\sqrt{\mathcal{L}(t_{j+1}, t_j)}$ for the cosine schedule and our learned schedule, finding that the increments approximately equalise over the diffusion path as expected by the discussion in Section 3.1. Figure 3 (right) shows the learned schedule spends more time at high-frequency details, we visualise a sampling trajectory in Figure 11.

Table 1: Sample quality measured by Fréchet Inception Distance (FID) versus schedule on CIFAR10 ($32 \times 32$), FFHQ, AFHQv2, ImageNet ($64 \times 64$). Pretrained models are used from Karras et al. (2022). All FIDs are calculated using 50000 samples. We highlight the best FID in **bold**. The ImageNet model lacks second-order differentiation, so no predictor optimised schedule is shown.

| Schedule | CIFAR-10 | FFHQ | AFHQv2 | ImageNet |
|---|---|---|---|---|
| Eq (22) $\rho = 3$ | 5.47 | 2.80 | **2.05** | 1.46 |
| Eq (22) $\rho = 7$ | **1.96** | 2.46 | **2.05** | **1.42** |
| LogLinear (Lu et al., 2022) | 2.05 | **2.42** | 2.06 | 1.45 |
| Convex Schedule | 22.1 | 2.43 | 2.48 | 1.64 |
| Corrector optimised | 1.99 | 2.46 | **2.05** | 1.44 |
| Predictor optimised | 1.99 | 2.48 | **2.05** | - |

| Schedules from low to high FID | FID | CO-Cost ($\times 10^3$) | PO-Cost ($\times 10^3$) | KLUB ($\times 10^6$) |
|---|---|---|---|---|
| Eq (22) $\rho = 7$ | **1.96** | 9.86 | 2.16 | 1.75 |
| CO (ours) | 1.99 | **9.54** | **2.09** | 1.39 |
| PO (ours) | 1.99 | 9.61 | **2.09** | 1.39 |
| LogLinear | 2.05 | 10.5 | 2.32 | 1.05 |
| Eq (22) $\rho = 3$ | 5.47 | 17.2 | 4.02 | 2.82 |
| Convex Schedule | 22.1 | 45.0 | 8.17 | **0.284** |

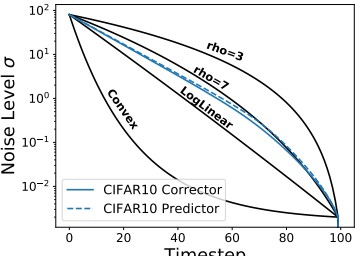

Figure 4: (**Left**) Costs associated with different schedule choices for the CIFAR10 dataset. Schedules are ordered from lowest FID to highest FID. We compare our Corrector-optimised (CO) cost and Predictor-optimised (PO) cost versus the Kullback-Leibler Upper Bound (KLUB) from Sabour et al. (2024). The minimum value for each cost is highlighted in **bold**. Note low cost is associated with low FID for our cost and not for the KLUB. (**Right**) Visualisation of schedules during generative sampling with 100 timesteps. "rho=3" and "rho=7" refer to Eq 22 with $\rho = 3$ and $\rho = 7$ respectively. LogLinear from Lu et al. (2022) and a convex schedule are also shown. We show our cost optimised schedules for CIFAR10 both using the corrector optimised cost and the predictor optimised cost.

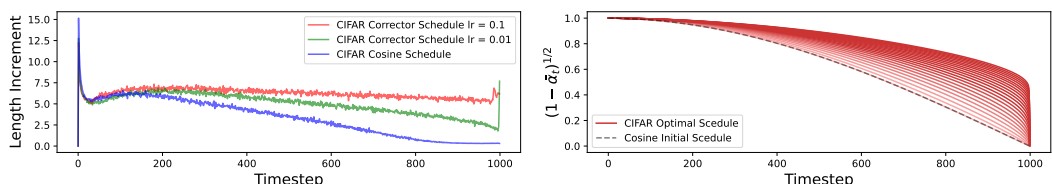

Figure 3: (**Left**) Incremental costs $\sqrt{\mathcal{L}(t_{j+1}, t_j)}$ for the cosine schedule and our online adaptive algorithm. Higher learning rates enforce equalisation of costs more quickly. (**Right**) Progression of the learned schedule during 40k training iterations, depicted through the standard-deviation $\sqrt{1 - \bar{\alpha}_t}$.

## 4.4 Sampling Pre-Trained Models

In this experiment we demonstrate that our algorithm can recover performant schedules for large image models used in practice and our schedules generate high quality samples. We use the pre-trained models from Karras et al. (2022), whose DDM is parameterised such that the forward noising distribution is of the form $p_{t|0}(x_t|x_0) = \mathcal{N}(x_t; x_0, \sigma_t^2 I)$. The scheduling problem then reduces to deciding on a stepping scheme through $\{\sigma_i\}_{i=1}^N$, $\sigma_N = 0$. Karras et al. (2022) suggest a polynomial based schedule with a parameter $\rho$ that controls the curvature of the schedule

$$\sigma_{i<N} = \left( \sigma_{\max}^{\frac{1}{\rho}} + \frac{i}{N-1} \left( \sigma_{\min}^{\frac{1}{\rho}} - \sigma_{\max}^{\frac{1}{\rho}} \right) \right)^{\rho} \quad \text{and} \quad \sigma_N = 0. \tag{22}$$

A lower $\rho$ value results in steps near $\sigma_{\min}$ being shortened and steps near $\sigma_{\max}$ being lengthened. Through analysing the truncation error for sampling in Karras et al. (2022), they find that setting $\rho = 3$ approximately equalises this error, however it is found empirically that $\rho = 7$ results in better sample quality. We also compare against a schedule that takes uniform steps in $\log \sigma$ space Lu et al. (2022) which we refer to as the LogLinear schedule and a schedule that takes a convex shape in log space. Schedule visualisations are provided in Figure 4 (right).

We sampled the pre-trained models using these schedules and computed the sample quality using FID. We use the same number of schedule steps (18 for CIFAR10, 40 for FFHQ and AFHQv2, 256 for ImageNet) and solver (Heun second order) as Karras et al. (2022). Our results are shown in Figure 4. Our optimised schedules are able to achieve competitive FID to the best performing $\rho = 7$ schedule hand-tuned in Karras et al. (2022). This is expected as our schedules take a similar shape to the $\rho = 7$ schedule as shown in Figure 4 (right). Therefore, our method provides an entirely automatic and hyperparameter free algorithm to recover this performant schedule that was previously only discovered through trial-and-error.

We further analyse how the number of discretisation points, $T$, used during sampling affects the quality of generated samples for different schedules. We report our results on CIFAR10 in Table 2. Notably, the FID decreases with $T$ for all schedules and achieves comparable FID once $T$ is large enough. However, when $T$ is small, only the optimised schedules maintain stable performance. This empirically demonstrates an optimised schedule can improve the sampling efficiency by allowing for coarser discretisations and, hence, faster sampling, as predicted by Theorem 3.1. We observe an identical trend for sFID in Table 3 in the Appendix C.2.

| # points, $T$ | 10 | 20 | 30 | 50 | 100 |
|---|---|---|---|---|---|
| CO (ours) | **2.46** | 2.02 | **2.04** | 2.06 | 2.07 |
| $\rho = 3$ | 50.75 | 3.92 | 2.09 | **2.01** | **2.05** |
| $\rho = 7$ | 2.70 | **2.00** | 2.06 | 2.05 | 2.07 |
| $\rho = 100$ | 3.09 | 2.06 | 2.05 | 2.06 | 2.07 |

Table 2: Comparison of FID across different amounts of discretisation points for different schedules on CIFAR10. CO stands for our corrector optimised schedule.

We also compare corrector optimised schedules to predictor optimised schedules in Table 1. They provide similar performance so, on image datasets, we encourage the use of the cheaper to compute corrector optimised schedule. Finally, in Figure 4 (left), we report the raw values of our corrector optimised costs and compare these costs to the values of the objective introduced in Sabour et al. (2024). Both algorithms aim to find schedules that minimise these costs and therefore it is desirable for low values of cost to be associated with good sample quality (i.e. low FID). We find that low values of our cost correlate much more closely with low FID than the objective introduced by Sabour et al. (2024). Indeed, Sabour et al. (2024) introduce a bespoke multi-stage optimisation for their cost because they found over-optimising their objective can lead to worse schedules which is explained by the objective not correlating well with FID. We further find that our predictor optimised costs are lower than the corrector optimised costs which is to be expected as the predictor reduces the work done by the corrector and thus reduces the incremental cost. The overall shape of schedule, however, between the corrector optimised and predictor optimised costs is similar.

## 5 Discussion

We have introduced a method for selecting an optimal DDM discretisation schedule by minimising a cost linked to the work done in transporting samples along the diffusion path. Our algorithm is computationally cheap and does not require hyperparameter tuning. Our learned schedule achieves competitive FID scores. Regarding limitations, the computation of $\mathcal{L}_p$ can be computationally expensive due the calculation of second derivatives, however, in Section 4.4 we found $\mathcal{L}_c$ to provide a cheap and performant alternative. Furthermore, our theory is derived assuming perfect score estimation. Future work can expand on the geometric interpretation of the diffusion path and links to information geometry to further refine the DDM methodology.

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

# A Analysis of incremental cost

## A.1 Proof of Theorem 2.1

*Proof.* We first note that by the mean value theorem, there exists $s(t, t') \in [t, t']$

$$\frac{\mathcal{L}(t, t')}{|t' - t|^2} = v(t')^2 \mathbb{E}_{X_t \sim p_t} \left[ \left\| \frac{\nabla G_{s(t,t')}(X_t)}{t' - t} \right\|^2 \right] \tag{23}$$

$$= v(t')^2 \mathbb{E}_{X_t \sim p_t} \left[ \left\| \nabla \dot{G}_{s(t,t')}(X_t) \right\|^2 \right] \tag{24}$$

Since all $t, t'$ we have $\|\nabla \dot{G}_s(t, t')(x)\|^2 \leq V(x)$, and $\mathbb{E}_{X_t \sim p_t}[V(X_t)] < \infty$, the dominated convergence theorem and the continuity of $v(t')$ implies,

$$\lim_{t' \mapsto t} \frac{\mathcal{L}(t, t')}{|t' - t|^2} = v(t)^2 \mathbb{E}_{X_t \sim p_t} \left[ \left\| \lim_{t' \mapsto t} \nabla \dot{G}_{s(t,t')}(X_t) \right\|^2 \right] \tag{25}$$

$$= v(t)^2 \mathbb{E}_{X_t \sim p_t} \left[ \left\| \nabla \dot{G}_{s(t,t')}(X_t) \right\|^2 \right]. \tag{26}$$

The last equality follows since $\nabla \dot{G}_t$ is continuous in $t$.

We will now compute $\dot{G}_t$. Since $F_{t,t}(x) = x$, we have $G_{t,t}(x) = 1$ for all $x$. This implies, we can express $\dot{G}_t$ in terms of the derivative of $\log G_{t,t'}$,

$$\dot{G}_t(x) = \left. \frac{\partial}{\partial t'} \log G_{t,t'}(x) \right|_{t'=t}, \tag{27}$$

where $\log G_{t,t'}$ equals,

$$\log G_{t,t'}(x) = \log p_{t'}(F_{t,t'}(x)) - p_t(x) + \log \det \nabla F_{t,t'}(x). \tag{28}$$

By combining Equation (27) to Equation (28), we obtain,

$$\dot{G}_t(x) = \left. \frac{\partial}{\partial t'} \log p_{t'}(F_{t,t'}(x)) \right|_{t'=t} + \left. \frac{\partial}{\partial t'} \log \det \nabla F_{t,t'}(x) \right|_{t'=t}. \tag{29}$$

For the first term in Equation (29), we use chain rule to obtain,

$$\frac{\partial}{\partial t'} \log p_{t'}(F_{t,t'}(x)) = \frac{1}{p_{t'}(F_{t,t'}(x))} \left( \frac{\partial p_{t'}}{\partial t'}(F_{t,t'}(x)) + \nabla p_{t'}(F_{t,t'}(x)) \cdot \frac{\partial F_{t,t'}}{\partial t'}(x) \right), \tag{30}$$

where '·' denotes a dot product of vectors. By evaluating at $t' = t$, we have

$$\left. \frac{\partial}{\partial t'} \log p_{t'}(F_{t,t'}(x)) \right|_{t'=t} = \frac{1}{p_t(x)} \frac{\partial p_t}{\partial t}(x) + \frac{1}{p_t(x)} \nabla p_t(x) \cdot \dot{F}_t(x) \tag{31}$$

$$= \frac{\partial}{\partial t} \log p_t(x) + \nabla \log p_t(x) \cdot \dot{F}_t(x). \tag{32}$$

For the second term in Equation (29), we note that as $\Delta t = t' - t \to 0$

$$F_{t,t'}(x) = x + \dot{F}_t(x) \Delta t + O(\Delta t^2). \tag{33}$$

This implies that the Jacobian determinant admits the following asymptotic expansion,

$$\log \det \nabla F_{t,t'} = \log \det(I + \nabla \dot{F}_t \Delta t + o(\Delta t)) \tag{34}$$

$$= \log(1 + \text{Tr} \nabla \dot{F}_t \Delta t + o(\Delta t)) \tag{35}$$

$$= \text{Tr} \nabla \dot{F}_t \Delta t + O(\Delta t^2). \tag{36}$$

Consequentially we have,

$$\left. \frac{\partial}{\partial t'} \log \det \nabla F_{t,t'}(x) \right|_{t'=t} = \text{Tr} \nabla \dot{F}_t(x). \tag{37}$$

By substituting in Equations (32) and (37) into Equation (29) we obtain,

$$\dot{G}_t(x) = \frac{\partial}{\partial t} \log p_t(x) + \nabla \log p_t(x) \cdot \dot{F}_t(x) + \text{Tr} \nabla \dot{F}_t(x). \tag{38}$$

$\square$

## A.2 Proof of Theorem 3.1

*Proof.* Let $s_i = i/T$ and $t_i = \varphi(s_i)$, Theorem 2.1 implies

$$\mathcal{L}(t_{i+1}, t_i) = \delta(t_{i+1})\Delta t_i^2 + o(\Delta_T^2), \tag{39}$$

$$\sqrt{\mathcal{L}(t_{i+1}, t_i)} = \sqrt{\delta(t_{i+1})}\Delta t_i + o(\Delta_T), \tag{40}$$

where $\Delta_T = \max_i |\Delta t_i|$. By the the mean value theorem, $\Delta_T \leq \sup_{s \in [0,1]} \dot{\varphi}(s)/T$ and hence is $O(T^{-1})$ as $T \to \infty$.

We will first establish the convergence of $\Lambda(\varphi, T)$. Using Equation (40) we obtain the following estimate for $T\mathcal{L}(\varphi, T)$,

$$\Lambda(\varphi, T) = \sum_{i=0}^{T-1} \sqrt{\mathcal{L}(t_{i+1}, t_i)} = \sum_{i=0}^{T-1} \sqrt{\delta(t_{i+1})}\Delta t_i + o(1). \tag{41}$$

In the limit as $T \to \infty$, this Riemann sum converges to $\Lambda$,

$$\lim_{T \to \infty} \Lambda(\varphi, T) = \int_0^1 \sqrt{\delta(t)}dt. \tag{42}$$

We will now obtain the limit of $T\mathcal{L}(\varphi, T)$. First denote $s_i = i/T$ and $\Delta s_i = 1/T$. Using the differentiability of $\varphi$ we have,

$$\Delta t_i = \varphi(s_{i+1}) - \varphi(s_i) = \frac{\dot{\varphi}(s_{i+1})}{T} + o\left(T^{-1}\right). \tag{43}$$

Substituting Equation (46) into Equation (39), we obtain the following estimate for $T\mathcal{L}(\varphi, T)$,

$$T\mathcal{L}(\varphi, T) = T\sum_{i=0}^{T-1} \mathcal{L}(t_{i+1}, t_i) \tag{44}$$

$$= T\sum_{i=0}^{T-1} \delta(t_{i+1})\Delta t_i^2 + o(T^{-1}) \tag{45}$$

$$= \sum_{i=0}^{T-1} \delta(\varphi(s_{i+1}))\dot{\varphi}(s_{i+1})^2 \Delta s_i + o(1). \tag{46}$$

In the limit as $t \to \infty$, this converges to the integral for $E(\varphi)$,

$$T\mathcal{L}(\varphi, T) = \int_0^1 \delta(\varphi(s))\dot{\varphi}(s)^2 ds = E(\varphi). \tag{47}$$

Combining Jensen's inequality with the fact that $\varphi$ is increasing implies,

$$E(\varphi) = \int_0^1 \delta(\varphi(s))\dot{\varphi}(s)^2 ds \geq \left(\int_0^1 \sqrt{\lambda(\varphi(s))}\dot{\varphi}(s)ds\right)^2 = \left(\int_0^1 \sqrt{\delta(t)}dt\right)^2 = \Lambda^2. \tag{48}$$

The last equality follows by substituting $t = \varphi(s)$. Note a schedule generator $\varphi^*$ obtains the Jensen lower bound if only if there is a $C$ such that for all $s \in [0, 1]$,

$$C = \delta(\varphi^*(s))\dot{\varphi}^*(s)^2. \tag{49}$$

By taking square roots and integrating from 0 to $s$,

$$\sqrt{C}s = \int_0^s \sqrt{\delta(\varphi^*(s'))}\dot{\varphi}^*(s')ds' = \int_0^{\varphi^*(s)} \sqrt{\delta(t)}dt = \Lambda(\varphi^*(s)). \tag{50}$$

By using the substitution in $s = 1$, along with the constraints $\Lambda(1) = \Lambda$ and $\varphi^*(1) = 1$ we obtain $\sqrt{C} = \Lambda$,

$$\sqrt{C} = \Lambda(\varphi^*(1)) = \Lambda(1) = \Lambda. \tag{51}$$

Finally, by inverting Equation (50), we conclude our proof,

$$\varphi^*(s) = \Lambda^{-1}(\sqrt{C}s) = \Lambda^{-1}(\Lambda s). \tag{52}$$

$\square$

### A.3 Comparison to Fisher Information

When $F_{t,t'}(x) = x$, the quantity $G_{t,t'}(x) = p_{t'}(x)/p_t(x)$ reduces to the Radon–Nykodym derivative between $p_{t'}$ and $p_t$, and $\dot{G}_t(x)$ reduce to the Fisher score function, $\frac{\partial}{\partial t} \log p_t(x)$. By Theorem 2.1 the corrector optimised cost satisfies, $\mathcal{L}_c(t, t') = \delta_c(t)\Delta t^2 + o(\Delta t)$, where

$$\delta_c(t) = v(t)^2 \mathbb{E}_{X_t \sim p_t} \left[ \left\| \nabla \dot{G}_t^2 \right\|^2 \right] = v(t)^2 \mathbb{E}_{X_t \sim p_t} \left[ \left\| \nabla \frac{\partial}{\partial t} \log p_t \right\|^2 \right]. \tag{53}$$

Suppose $p_t$ is sufficiently regular, and the Poincaré inequality (Poincaré, 1890) holds. Since the expectation of the Fisher score with respect to $p_t$ is zero, we have,

$$\delta_c(t) \geq C(t)v(t)\mathbb{E}_{X_t \sim p_t} \left[ \left( \frac{\partial}{\partial t} \log p_t \right)^2 \right] = C(t)v(t)\delta_F(t), \tag{54}$$

where $C(t) > 0$ is some constant. Here $\delta_F(t) = \text{Var}_{X_t \sim p_t} \left[ \frac{\partial}{\partial t} \log p_t \right]$ is the Fisher information for the diffusion path $p_t$. Suppose we view the diffusion path as a curve in the probability distribution space with metric $\delta_c(t)$. Equation (54) shows that the topology induced by $\delta_c(t)$ is stronger than the Fisher information, and geometry is more regular.

## B Training Algorithms

### B.1 Adaptive training

We may incorporate Algorithm 1 into an online algorithm used during training. For a fixed score function along the diffusion path, there is an optimal schedule. Similarly, for a fixed schedule, there is an optimal score function that can be learnt from the data. To incorporate these two steps, we propose a two-step algorithm for online training.

First, for a fixed schedule, we optimise the score function. Then, using this estimated score function, we compute the optimal schedule through Algorithm 1. To add regularity throughout training, as our score predictions are over batches rather than the entire dataset, we do not replace the current schedule with our computed optimal one. Instead, we take a weighted combination of the current schedule with the computed optimal one.

The weighting factor $\gamma \in (0, 1)$ is akin to a learning rate for the schedule optimisation. If $\gamma$ is set too high, our schedule learning may be overly influenced by the current batch, which could negatively affect the score training performance.

---

**Algorithm 2** `AdaptiveScheduleTraining`

---

**Require:** Initial schedule $\mathcal{T} = \{t_i\}_{i=0}^T$, learning rate $\gamma \in (0, 1)$, score estimate $s_\theta$
1: **while** not converged **do**
2:    **for** each batch $B$ from data **do**
3:       Fix $\mathcal{T}$ and assign $\theta \leftarrow \text{argmin}_\theta \mathcal{L}_{\text{training}}(\theta, B, \mathcal{T})$
4:       Fix $s_{\theta^*}$ and over batch estimate $\mathcal{L}(t_{i+1}, t_i)$,   $i = 0, \ldots, T-1$     (Equation (13))
5:       Assign $\mathcal{T}^* \leftarrow \text{UpdateSchedule}(\mathcal{T}, \mathcal{L}(t_{i+1}, t_i))$
6:       Update time locations $t_i \leftarrow \gamma t_i^* + (1 - \gamma)t_i$
7:    **end for**
8: **end while**

---

### B.2 Estimating predictor optimised cost

**Proposition B.1.** *Let $F_{t,t'}$ be the predictor map given by the forward Euler discretisation* (8) *of the probability flow ODE. For $N \in \mathbb{N}$ and let $\hat{J}_{t,N}(x)$ to be the Jacobian of the Hutchinson trace estimator (Hutchinson, 1989) for $\nabla(\nabla \log p_t(x))$ at $x \in \mathbb{X}$ and $t \in [0, 1]$,*

$$\hat{J}_{t,N}(x) = \frac{1}{N} \sum_{n=1}^N \nabla(v_n^T J_t(x)v_n), \quad v_n \sim \mathcal{N}(0, I). \tag{55}$$

*If $\Delta t$ is small enough such that,*

$$\Delta t \text{Tr} \left( f(t)I - \frac{1}{2}g(t)^2 \nabla^2 \log p_t(x) \right) < 1. \tag{56}$$

*Then, as $N \to \infty$ the following limit exists almost surely,*

$$\nabla \log \det \nabla F_{t,t'}(x) = -\frac{\Delta t}{2}g(t)^2 \lim_{N \to \infty} \hat{J}_{t,N} + O(\Delta t^2). \tag{57}$$

*Proof.* The probability flow ODE update is given through

$$F_{t,t'}(x) = x + \Delta t \left( f(x)x - \frac{1}{2}g(t)^2 \nabla \log p_t(x) \right). \tag{58}$$

By taking the gradient, we have,

$$\nabla F_{t,t'}(x) = I + \Delta t \left( f(t)I - \frac{1}{2}g(t)\nabla^2 \log p_t(x) \right). \tag{59}$$

Since $\log \det(1 + \Delta t) = \Delta t \text{Tr}A + O(\Delta t^2)$ that holds when $\text{Tr}(A) < 1$, our assumption in Equation (56) hold, we can apply a Taylor expansion

$$\log \det \nabla F_{t,t'}(x) = \Delta t \text{Tr} \left( f(t)I - \frac{1}{2}g(t)\nabla^2 \log p_t(x) \right) + O(\Delta t^2). \tag{60}$$

By taking a gradient, we have

$$\nabla \log \det \nabla F_{t,t'}(x) = -\frac{\Delta t}{2}g(t)^2 \nabla \text{Tr} \left( \nabla^2 \log p_t(X) \right) + O(\Delta t^2) \tag{61}$$

$$= -\mathbb{E}(vv^T)\frac{\Delta t}{2}g(t)^2 \nabla \text{Tr} \left( \nabla^2 \log p_t(X) \right) + O(\Delta t^2) \tag{62}$$

$$= -\frac{\Delta t}{2}g(t)^2 \mathbb{E} \, \nabla \text{Tr} \left( v^T \nabla^2 \log p_t(X)v \right) + O(\Delta t^2). \tag{63}$$

The last line used the fact that the Hessian of $\log p_t$ is equivalently the Jacobian $\nabla(\nabla \log p_t)$, the latter we can approximate unbiasedly using the Hutchinson trace estimator to gain $\hat{J}_{t,N}(x)$. $\quad\square$

### B.3 Denoising Score Matching Weighting

The standard denoising score matching weighting used by Song et al. (2021) is $\lambda(t) \propto 1/\mathbb{E}[||\nabla \log p_{t|0}(X_t|X_0)||^2]$. For $p_{t|0}(x_t|x_0) = \mathcal{N}(x_t; s(t)x_0, \sigma^2(t)I)$ we have,

$$\log p_{t|0}(x_t|x_0) = -\frac{1}{2}\frac{||x_t - s(t)x_0||^2}{\sigma(t)^2} - \frac{d}{2}\log\left(2\pi\sigma(t)^2\right). \tag{64}$$

Therefore,

$$\nabla_{x_t} \log p_{t|0}(x_t|x_0) = \frac{s(t)x_0 - x_t}{\sigma(t)^2} \tag{65}$$

The expectation for $\mathbb{E}[||\nabla \log p_{t|0}(X_t|X_0)||^2]$ is taken with respect to $X_t \sim p_{t|0}$. We therefore have,

$$\mathbb{E}[||\nabla \log p_{t|0}(X_t|X_0)||^2] = \mathbb{E}_{p_{t|0}(X_t|X_0)} \left[ \left\| \frac{s(t)X_0 - X_t}{\sigma(t)^2} \right\|^2 \right] \tag{66}$$

$$= \mathbb{E}_{\mathcal{N}(\epsilon;0,I)} \left[ \left\| \frac{s(t)X_0 - s(t)X_0 - \sigma(t)\epsilon}{\sigma(t)^2} \right\|^2 \right] \tag{67}$$

$$= \mathbb{E}_{\mathcal{N}(\epsilon;0,I)} \left[ \left\| \frac{-\epsilon}{\sigma(t)} \right\|^2 \right] \tag{68}$$

$$= \frac{1}{\sigma(t)^2}\mathbb{E}_{\mathcal{N}(\epsilon;0,I)} \left[ ||\epsilon||^2 \right] \tag{69}$$

$$= \frac{d}{\sigma(t)^2}. \tag{70}$$

where on the second line we have used the reparameterisation trick. Therefore, we have that the standard weighting for denoising score matching is $\lambda(t) \propto \sigma(t)^2$.

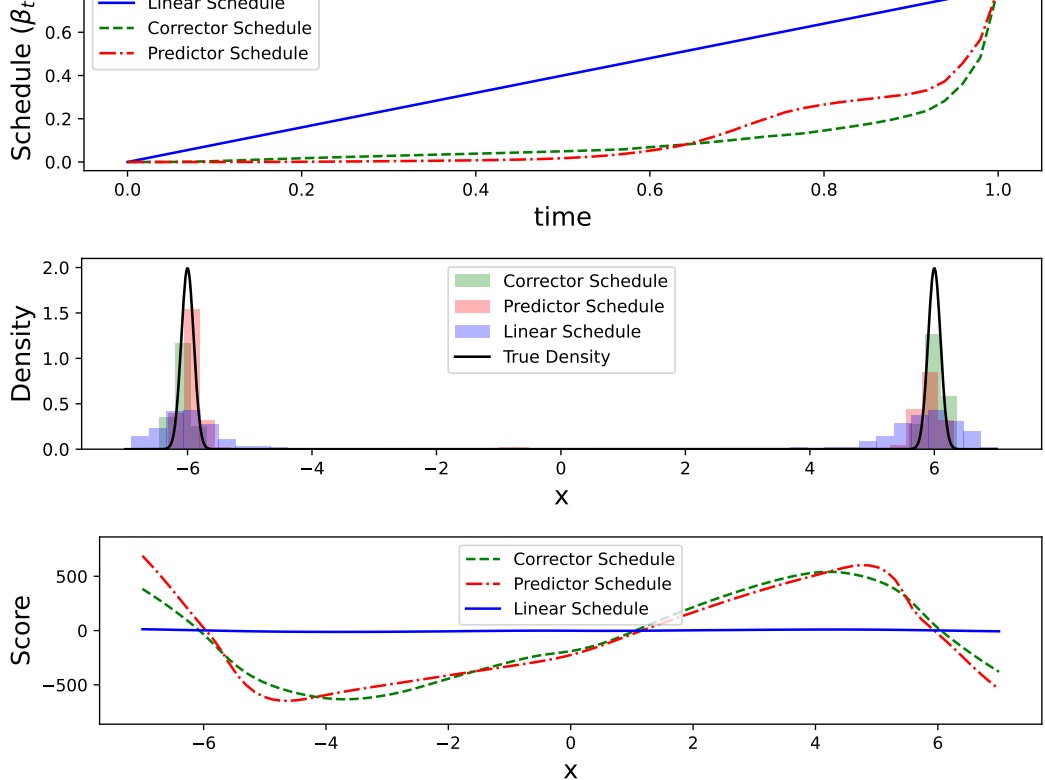

Figure 5: Schedules and density estimates for: linear (blue); Stein score optimised (green); and predictor optimised (red) schedules. The predictor optimised schedule identifies a bump along the diffusion path where the reference Gaussian density splits into two modes. In the regions where the score is evaluated (around $\pm 6$), our trained score is accurate compared to the linear schedule score, which fails to match the slope of the true score, resulting in a wider variance density estimate.

## C    Experiment Details

### C.1    1D Density Estimation

For all 1D experiments, we train a one spatial dimension model with continuous time encoding via Gaussian Fourier features to embed time values into a higher-dimensional space. The model architecture includes a hidden dimension of 128, five layers, an embedding dimension of 12, and one residual time step. It features a combination of residual blocks, both incorporating linear layers with GELU activation and LayerNorm, tailored to integrate time embeddings.

#### C.1.1    Bimodal Example

We train our model in the form of $f(t) = \beta_t/2$ and $g(t) = \sqrt{\beta_t}$ using Algorithm 2 with $\gamma = 0.1$. We use the weight function $v_{t_n}^2 = (1 - \bar{\alpha}_{t_n})$, where $\alpha_{t_n} = 1 - \beta_{t_n}$ and $\bar{\alpha}_{t_n} = \prod_{i=1}^{n} \alpha_{t_i}$. We train our model for 5 thousand iterations using both a fixed linear schedule and our optimisation algorithm initialised at the linear schedule. Due to the non-linear dependence during training of the transport-optimised schedule, initialisation in this context plays an important role. We initialise our predictor-optimised schedule with the optimal schedule generated with respect to $\mathcal{L}_c$ without the Jacobian term as a first approximation. We then add the Jacobian term, optimise the schedule, and train for an additional 5 thousand iterations.

In Figure 5 we see that the linear schedule forms an estimate with greater variance than the true data and the predicted densities of the optimised schedules.

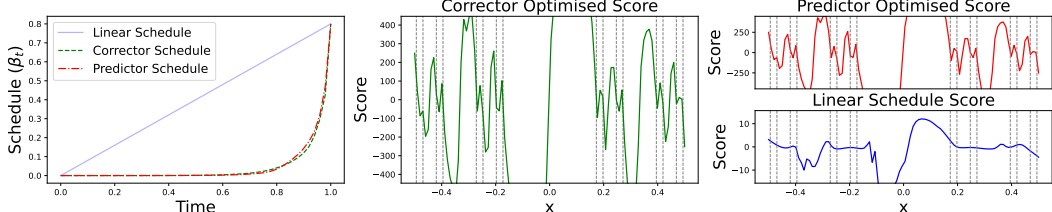

Figure 6: Comparison of sampling from a mollified Cantor distribution using DDMs with two different schedules: linear (blue) and optimised (green). The optimised schedule enables the DDM to capture eight distinct data modes centered on the mollified Cantor distribution (grey), whereas the linear schedule does not have clear mode separation. The optimised schedule diffusion accurately predicts the score for the mollified Cantor distribution, being near vertical lines interweaving the Cantor set, where the linear schedule fails to adequately approximate the score.

### C.1.2 Mollified Cantor Distribution

We train our model with $f(t) = \beta_t/2$ and $g(t) = \sqrt{\beta_t}$ using the online Algorithm 2 with $\gamma = 0.01$. The weight function is $v_{t_n}^2 = (1 - \bar{\alpha}_{t_n})$, where $\alpha_{t_n} = 1 - \beta_{t_n}$ and $\bar{\alpha}_{t_n} = \prod_{i=1}^{n} \alpha_{t_i}$. To capture high-frequency details, we train the one-dimensional model for 150,000 iterations using both a fixed linear schedule and our optimisation algorithm initialised at the linear schedule. The difference in sample quality is evident in Figure 1.

### C.2 Pretrained Image Models

For sampling pretrained image models, we use the networks and implementation from Karras et al. (2022), `https://github.com/NVlabs/edm`. We compute the FID using the provided FID script within the codebase with the standard $50,000$ samples using the same fixed seeds $0 - 49999$ for all schedules. For each image dataset, we use the default sampling strategy included in the codebase by Karras et al. (2022). For unconditional CIFAR10 this is a deterministic 2nd order Heun solver with 18 timesteps, therefore a total of 35 NFE (number of function evaluations) for the underlying denoising network. For both unconditional FFHQ and unconditional AFHQv2 the same deterministic 2nd order Heun solver is used with 40 timesteps (NFE = 79). For class conditional ImageNet, the bespoke stochastic solver from Karras et al. (2022) is used with 256 timesteps (NFE=511). The stochasticity settings are left at their default values for this dataset of $S_{\text{churn}} = 40$, $S_{\text{min}} = 0.05$, $S_{\text{max}} = 50$, $S_{\text{noise}} = 1.003$.

To compute the corrector optimised schedule for each dataset, we use Algorithm 1. We use 100 data samples for each dataset when computing $\mathcal{L}_c$ as we find the variation in learned schedule is small between different samples from the dataset. Our initial discretisation schedule used to calculate $\Lambda(t)$ is LogLinear with 100 steps. We then fit a monotonic spline to the cumulative estimated $\Lambda(t)$ and invert this function to find $\varphi^*$ function from which we can derive our schedules. This takes on the order of 5 minutes to find the Corrector Optimised Schedule for CIFAR10 on a single RTX 2080Ti GPU. For predictor optimised schedules we repeat the same procedure however also include estimation of the Hessian term. We use 5 samples of $v$ for each image datapoint when using Hutchinson's trace estimator. It takes 5 GPU hours to compute the Predictor Optimised Schedule for CIFAR10 due to the extra computational cost of computing the second derivatives.

We compare the corrector optimised and predictor optimised schedules for the 4 image datasets in Figure 8. We find that all of the schedules have the same general shape with increasing step sizes in log space as the generative process approaches clean data. However, the curvature of the schedule varies between datasets which is to be expected as the schedule is determined through the score which will vary depending on the data distribution. We find that, in general, the higher resolution datasets (FFHQ, AFHQv2 and ImageNet) favour shorter steps nearer the start of the generative process at the expense of larger steps at low noise levels near the end of the generative process.

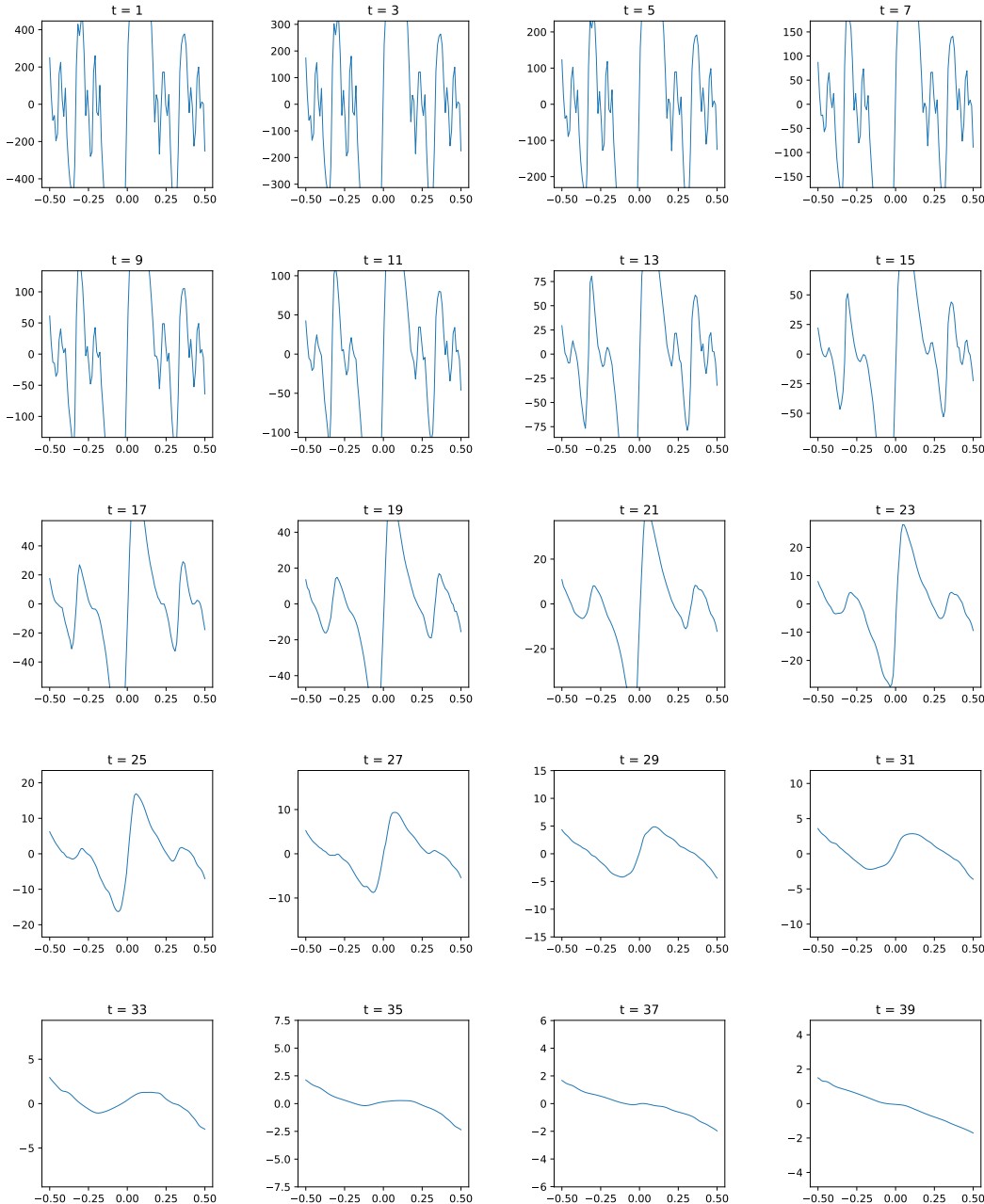

Figure 7: Evolution of the estimated score for the mollified Cantor distribution Section 4.1 with a Corrector Optimised Schedule. In this case the linear schedule fails to evenly progress the progression of the score, see Figure 6 showing the terminal score estimate in this case. The estimated score exhibits a self-similar nature of interweaving roots around the centers of mass of the mollified Cantor distribution. Identification of these roots amounts to estimated modes in our density estimate, see Figure 1.

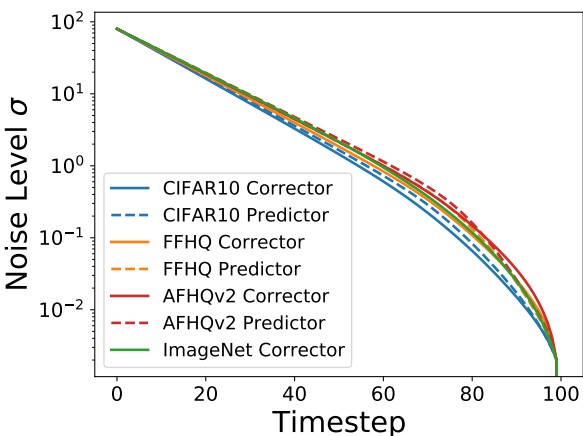

Figure 8: Corrector optimised and predictor optimised schedules for the 4 image datasets, CIFAR10, FFHQ, AFHQv2 and ImageNet.

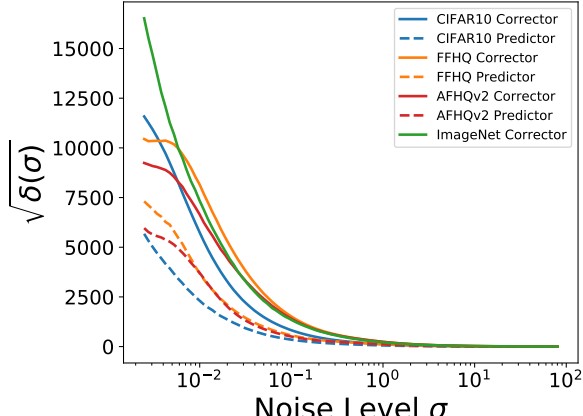

Figure 9: Local cost $\sqrt{\delta(\sigma)}$ versus $\sigma$ for 4 image datasets and using the corrector optimised versus predictor optimised costs.

We analyse the local cost as a function of noise level $\sqrt{\delta(\sigma)}$ for the 4 images datasets within Figure 9. This local cost is used as the metric when determining velocities in the space of schedules from $p_1$ to $p_0$.

| # points, $T$ | **10** | **20** | **30** | **50** | **100** |
|---|---|---|---|---|---|
| CO (ours) | 3.94 | 3.78 | 3.80 | 3.81 | 3.81 |
| $\rho = 3$ | 24.08 | 4.90 | 3.80 | 3.77 | 3.80 |
| $\rho = 7$ | 4.02 | 3.76 | 3.78 | 3.80 | 3.81 |
| $\rho = 100$ | 4.31 | 3.81 | 3.81 | 3.81 | 3.81 |

Table 3: Comparison of sFID across different amounts of discretisation points for different schedules on CIFAR10. CO stands for our corrector optimised schedule.

### C.3 Online Schedule Optimisation of Images

**MNIST**: For the MNIST experiments, we trained a model with an image size of 32, 32 channels, with a U-Net architecture, with 1 residual block per U-Net resolution, without learning sigma, and

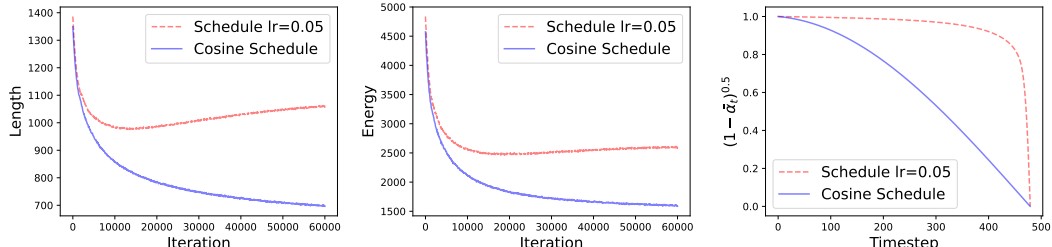

Figure 10: Progression of the length and energy Equation (18) over training of MNIST. Both models are trained from initialisation, one with adaptive schedule learning (red) and one without (blue). We can see that the energy and length quantities increase during training. Recall that for a fixed path of scores $t \mapsto \nabla \log p_t$ that the length $\Lambda$ is constant. As we are learning the score, this value is *not* constant during training. Interestingly, by optimising the schedule during training we observe a larger length value, possibly indicating that the diffusion path learned with the optimised schedule differs greatly from the path learned without.

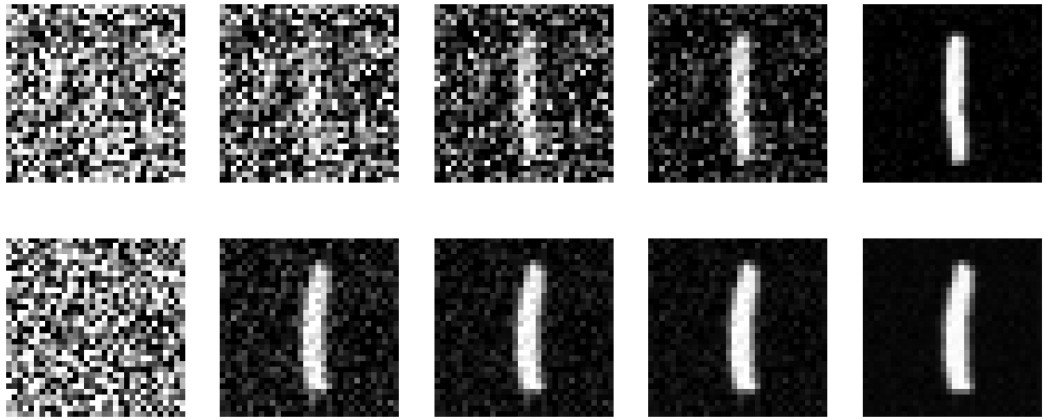

Figure 11: Sample progression of MNIST digits for the standard cosine schedule with $\epsilon = 0.008$ (top) against our optimised schedule (bottom). As we can see, the cosine schedule spends more time near the Gaussian reference distribution whereas the optimised schedule quickly determines large scale features and spends more time toward the data distribution.

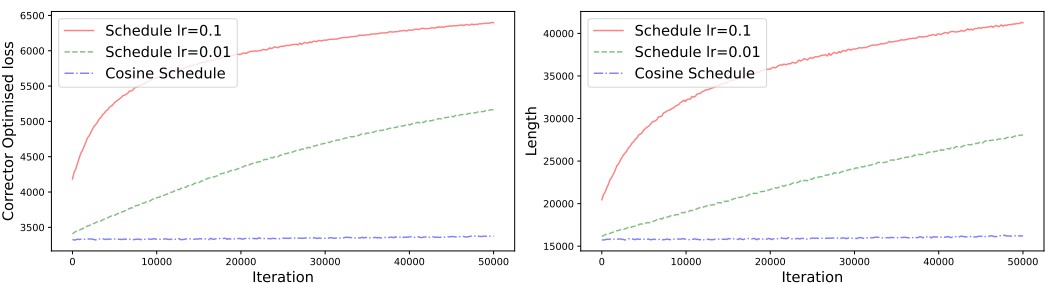

Figure 12: Progression of the length $\Lambda$ and cost through online training for CIFAR-10. For the larger learning rate, Algorithm 2 seems to garner a larger $\Lambda$ value at a faster rate that the lower schedule learning rate. For the fixed cosine schedule run, $\Lambda$ is stable, perhaps because the score estimate has stabilised for the fixed cosine schedule already during the model training burn-in phase.

with a dropout rate of 0.3. The diffusion process was configured with 500 diffusion steps. Training was conducted with a learning rate of 1e-4.

The schedule was also initialised with the Cosine schedule and trained for 60,000 iterations on an NVIDIA 1080 Ti GPU with 12 GB RAM. The batch size for MNIST was set to 128. We trained two models: one with the schedule optimisation and one without. The schedule training rate was set to $\gamma = 0.05$, as stated in Algorithm 2. Training took approximately 12 hours for either model.

**CIFAR-10**: For the CIFAR-10 experiments, we trained a model with an image size of 32, 128 channels, and a U-Net architecture with 3 residual blocks per multiplier resolution (described in codebase Nichol and Dhariwal (2021)), without learning sigma, and with a dropout rate of 0.3. The diffusion process was configured with 1000 diffusion steps and a cosine noise schedule.

The schedule was initialised with the Cosine schedule and trained for 160,000 iterations on 4 NVIDIA A40 GPUs, each with 48 GB RAM. The batch size was set to 1,536. After training for 160,000 iterations, we trained two models online: one with the corrector schedule optimisation and one without, for an additional 50,000 iterations on a single GPU with a batch size of 384. The burn in phase for training took approximately 50 hours, with the individual schedule optimisations after this taking approximately 24 hours.

### C.4 Licenses

Codebases:

- Improved Denoising Diffusion Probabilistic Models Nichol and Dhariwal (2021): MIT License
- Elucidating the Design Space of Diffusion-Based Generative Models Karras et al. (2022): Attribution-NonCommercial-ShareAlike 4.0 International

Datasets:

- CIFAR-10 Krizhevsky et al. (2009): MIT license
- FFHQ Karras et al. (2018): Creative Commons BY-NC-SA 4.0 license
- AFHQv2 Choi et al. (2020): Creative Commons BY-NC 4.0 license
- ImageNet Deng et al. (2009): Unknown License

## D   Acknowledgements of Funding

CW acknowledges support from DST Australia. AC acknowledges support from the EPSRC CDT in Modern Statistics and Statistical Machine Learning (EP/S023151/1). SS acknowledges support from the NSERC Postdoctoral Fellow Program.

