# OpenReview forum: "Score-Optimal Diffusion Schedules"
_NeurIPS.cc/2024/Conference — NeurIPS 2024 poster_

### Official Review · Reviewer_g4tL · 2024-06-20

**Soundness:** 4
**Presentation:** 4
**Contribution:** 4
**Rating:** 7
**Confidence:** 4

**Summary:**

This paper proposes a novel algorithm for adaptively selecting an optimal discretisation schedule in the training of diffusion model. The proposed method does not require hyper-parameter tuning and adapts to the dynamics and geometry of the diffusion path. This method also achieves competitive FID scores on image datasets.

**Strengths:**

+ The findings of the paper are interesting. Current schedule designs are heuristic, and this paper provides a theoretical analysis, which is of interest to the community.
+ The proposed online data-dependent adaptive discretization schedule tuning algorithm is simple and effective.
+ The mathematical analysis presented in this paper is exceptionally detailed. Meanwhile, adequate related work was discussed in detail. The authors also provide corresponding code that makes the work in this paper very solid.
+ The authors provide clear visual comparisons that show the distribution of optimized times in detail.

**Weaknesses:**

+ The experiments in this paper are somewhat weak and lack validation for the LDM model. In fact, the LDM model is widely used, so it is important to validate the proposed method for LDM.
+ Some other diffusion frameworks also use schedule discretization in the same way, e.g., VP-SDE [1] and Flow Matching [2], is the method proposed in this paper robust for them as well?

[1] Score-Based Generative Modeling through Stochastic Differential Equations ICLR 2021

[2] Flow Matching for Generative Modeling ICLR 2023

+ The authors should provide additional quantitative and visual comparisons, as the FID metric alone is insufficient for assessing generation quality [1]. Low FID scores do not necessarily correlate with superior generative performance. Therefore, incorporating other metrics such as sFID, Recall, and Precision would offer a more comprehensive evaluation. This enhanced analysis will help further substantiate the validity of the proposed method.

[1] The Role of ImageNet Classes in Fréchet Inception Distance ICLR 2023

[2] Diffusion Models Beat GANs on Image Synthesis  NeurIPS 2021

+ The mathematical parts of this paper are somewhat difficult to understand, especially the differential geometry part of it, and I would suggest that the authors add some background and notation notes.

**Questions:**

Please see Weakness.

**Limitations:**

Please see Weakness.

---

> ### Author Rebuttal · Authors · 2024-08-07
>
> We thank the reviewer for their kind comments on the strengths of our paper. We believe that our schedule tuning algorithm could serve as a strong default for diffusion models and might be applicable more broadly, though a more extensive exploration is difficult to undertake in an initial paper. We hope to use this work as a starting point for further investigation into general schedule tuning in generative models.
>
> In the context of diffusion models, we welcome the reviewer's suggestions for more comprehensive numerical validation. To this end, we have presented preliminary summary statistics for sFID. We are appreciative of the reviewers suggestions for additional metrics, and agree that their addition to a potential camera ready article would increase the strength of the paper.
>
> We believe that our methodology could be applied to flow-matching based models, but this would also require mild adjustment to the analysis we present. We hope our present work could be foundational to a potential future study.
>
> We agree that validation of LDM with our method is a worthwhile and meaningful venture. To this end, we have implemented our method in the HuggingFace LDM library, but due to time constraints do not currently have FID to report. In a potential camera ready article, we would additionally include FID and sFID comparisons on optimised and standard schedules for Latent Diffusion Models.
>
> _The authors should provide additional quantitative and visual comparisons, as the FID metric alone is insufficient for assessing generation quality [1]. Low FID scores do not necessarily correlate with superior generative performance. Therefore, incorporating other metrics such as sFID, Recall, and Precision would offer a more comprehensive evaluation. This enhanced analysis will help further substantiate the validity of the proposed method._
>
> We thank the reviewer for their suggestion. We would like some assistance from the reviewer, we would like to confirm that sFID refers to the metric computed in:
>
> Nash C, Menick J, Dieleman S, Battaglia PW. Generating images with sparse representations. arXiv preprint arXiv:2103.03841. 2021 Mar 5.
>
> We have implemented this metric and computed preliminary results for sFID on the CIFAR dataset which are included in our rebuttal document. We have found that the results for sFID correlate strongly with the results for FID in this case. In a potential camera-ready version, we would incorporate this metric across all the datasets presented. We appreciate the reviewer’s input, which has helped substantiate the validity of our method and improve the results presented in our paper.
>
> _The mathematical parts of this paper are somewhat difficult to understand, especially the differential geometry part of it, and I would suggest that the authors add some background and notation notes._
>
>  We agree with the reviewer that this mathematical content is new to the diffusion literature. In a potential camera-ready version, we would include a brief introduction to metric tensors and charts commonly used in differential geometry, along with their relation to the work we present here. We appreciate the reviewer’s suggestion, which will help us present our work more clearly.

---

> > ### Comment · Reviewer_g4tL · 2024-08-08
> >
> > I appreciate the author's thorough response to my questions, which are satisfactorily addressed. I find the work to be meaningful and have accordingly increased my score.

---

> > > ### Author Response · Authors · 2024-08-11
> > >
> > > We thank the reviewer for assessing our work and further thank them for increasing their score. Thank-you!

---

### Official Review · Reviewer_8ZVm · 2024-07-06

**Soundness:** 2
**Presentation:** 1
**Contribution:** 2
**Rating:** 4
**Confidence:** 3

**Summary:**

This paper proposes a method to optimize for the discretization schedule of the diffusion sampling process. The main idea is to minimize a surrogate for the total length of the diffusion sampling path between two consecutive time steps, where the length is defined in terms of the Fisher divergence between two steps. Experiments show that the proposed scheme can find schedules competitive with the ones discovered with manual search.

**Strengths:**

* The problem studied is important for diffusion generative modeling as the manual search for a good sampling schedule can be time-consuming.
* The proposed method is simple, despite the rather convolutated presentation.
* Empirical evaluation shows that the proposed scheme can have on-par performance with hand-tuned schedules.

**Weaknesses:**

* The presentation of the paper could use a lot of improvement. I found many sections unnecessarily convoluted without conveying much insight.
  - The whole setup of predictor/corrector seems unnecessary to me. If I understand correctly, Section 2.1 and 2.2 are used to introduce the cost which turns out to be just the Fisher divergence either between $p_{t_{i+1}}$ and $p_{t_{i}}$ (the so-called corrector optimized cost) or $F_\sharp p_{t_i+1}$ and $p_{t_i}$ for $F$ equal to the one-step of ODE integration.
  - Section 2.3 and Theorem 2.1 do not seem to be related to the actual algorithm.
* The theoretical insights in Section 3 are unclear to me. Section 3.1 seems to be regurgitating the fact in differential geometry that the energy-minimizing path connecting two points is a constant-speed geodesic. In particular, I'm unable to identify interesting theoretical contributions related to the setting of diffusion generative modeling.
* The computation cost could be quite high, especially using the predictor-optimized schedule, which does not really offer a noticeable advantage over the cheaper corrector-optimized schedule. This seems to suggest one should always choose the simpler corrector-optimized schedule. Yet with the corrector optimized schedule, the picture of predictor/corrector seems irrelevant since the predictor is just the identity.

**Questions:**

* Line 95, why is the Markov kernel taking in a probability measure as the second argument, as opposed to the usual definition where it takes in a Borel set (so that conditioned on the first argument you get a probability measure)?
* Line 136, what is the point of introducing coupled noises in (9) and (10) if they are going to be canceled anyway?
* Line 218, what is the "distance" referred to here?
* Line 235, can the authors comment on why we know $\Delta t$ is sufficiently small a priori (since we are optimizing the time intervals themselves)?
* Can the authors justify the usage of simple cubic interpolation in step 3 of Algorithm 1 in order to invert $\Lambda$? Why would interpolation work well?



Minor comments:
* In (3), why is there a tilde on $W_t$?
* Line 75, what are "Lebesgue probability measures"?
* In (4), one could split (3) into a prediction and a correction term differently. Why should this specific way of splitting?

**Limitations:**

Limitations are discussed.

---

> ### Author Rebuttal · Authors · 2024-08-07
>
> We are grateful for the reviewers for their constructive feedback. We will address each point raised individually below:
>
>  - _The whole setup of predictor/corrector seems unnecessary to me._
>
> See general rebuttal response.
>
> - _Section 2.3 and Theorem 2.1 do not seem to be related to the actual algorithm._
>
> Section 2.3 and Theorem 2.1 are necessary links that take us from the Fisher Divergence at two time points t, t’ to a metric that can be used to derive the optimal schedule. In Section 3.1 when we come to find the costs associated with diffusion paths, we require an infinitesimal notion of cost that can be integrated along the diffusion path. The Fisher Divergence as stated in Section 2.2 is unsuitable for this purpose because it is evaluated between two times t and t’. In Section 2.3 and Theorem 2.1 we derive this corresponding infinitesimal cost. Our algorithm relies entirely on the approximation of this infinitesimal cost along the diffusion path.
> We appreciate the reviewer highlighting this lack of clarity in the significance of Section 2.3 and Theorem 2.1. In a revised version we will add a clarifying sentence before Theorem 2.1 to emphasise its relevance and importance.
>
> - _Section 3.1 seems to be regurgitating the fact in differential geometry that the energy-minimizing path connecting two points is a constant-speed geodesic_
>
> Our algorithm is grounded in the theoretical insights from Section 3. The relevance of Section 3.1 lies in demonstrating how schedule optimization in diffusion models can be reframed as finding an energy-minimising path with an appropriate metric.  The key connection between differential geometry and diffusion models is established in Theorem 2.1, where we derive the Reimannian metric compatible with the objectives of diffusion models, and demonstrate how to approximate the energy using the Fisher divergence. The material from differential geometry included in this section is meant to ensure completeness. To improve the clarity of this section, we will add a linking sentence that explicitly states how the local metric tensor from Theorem 2.1 enables us to use differential geometry to study diffusion schedule design.
>
> - _The computation cost could be quite high.._
>
> See general rebuttal response.
>
> Questions:
>
> - _Line 95_
>
> Thank you for pointing out this typo.
>
> - _Line 136_
>
> The introduction of coupled noises in Section 2.2 is a direct consequence of performing a perturbation analysis on the Langevin dynamics trajectories. Specifically, we are interested in understanding how these trajectories evolve over a small time interval when the score function is slightly perturbed.
>
> By coupling the noise terms in equations (9) and (10), we ensure that any observed differences between these trajectories are solely due to the perturbation and not due to random fluctuations. The noise cancels out naturally because we have isolated the perturbation of the score; this cancellation is a result of the computational approach rather than a manufactured effect. Without coupling the noises, the comparison would be less informative, as it would be unclear whether differences were due to the perturbation or simply random noise.
>
> - _Line 218_
>
> Here "distance" is intended to mean "length along the path," as referenced in the preceding lines 202-207. We will update the text to more clearly state "length along the path".
>
> - _Line 235_
>
> We have not claimed that we know $\Delta t$ is sufficiently small, but rather if it is small, then our approximation is valid. One way to assess if a grid is fine enough is by computing the statistical length $\Lambda$, which is independent of the schedule, across a range of grid sizes. When this value stabilises for a particular $\Delta t$, it indicates that the desired integral approximation of Equation 21 has been achieved.
>
> - _Simple cubic interpolation in step 3_
>
> We also remark that be apprimae $\Lambda^{-1}(t)$ directly by interpolating with knots $\{(\Lambda(t_i), t_i)\}$. This avoids needing to approximate $\Lambda(t)$ and numerically inverting. Since we avoid numerical inversion methods, the choice of interpolant is not overly critical. We use cubic interpolation as an example, but any interpolation method respecting monotonicity (e.g. linear) can also be used. We will add this comment after Algorithm 1.
>
> - _In (3)_
>
> In the context of the forward/backward SDEs in a diffusion model, the Brownian increments for the forward and backward SDEs are independent of one another. We use a tilde to emphasise that the Brownian increment for the forward diffusion ($\tilde{W}$) is different from the Brownian increment of the backward diffusion ($W$).
>
> - _Line 75_
>
> Lebesgue probability measures are those that have a density function with respect to the Lebesgue measure. We've included standard references to make this clearer for readers who may be unfamiliar with probability theory and thank the reviewer for improving the accessibility of our work.
>
> - _In (4)_
>
> This specific way of splitting is chosen to identify the unique ODE that transports samples from one density $p_{t}$ along the diffusion path to $p_{t'}$. This approach was originally identified and used in the seminal work of Yang Song [reference]. We will add a reference in the text to where this splitting is derived in Yang Song's paper.

---

> > ### Comment · Reviewer_8ZVm · 2024-08-11
> > **Response to authors**
> >
> > Thank you for the detailed response. I remain unconvinced the whole predictor/corrector setup is necessary. For instance, (13) makes sense to me at first glance (as Fisher divergence), and the predictor/corrector setup only adds more confusion. In addition, the authors acknowledged that there is no real benefit from the predictor-optimized schedule as opposed to the simpler corrector-optimized schedule where there is essentially no predictor. The novelty and significance of Section 3.1 remains unclear to me. My other concern about the justification of the ad-hoc way of inverting $\Lambda(t)$ is also not well addressed. As such, I would like to keep my score unchanged.

---

> > > ### Author Response · Authors · 2024-08-11
> > >
> > > Thank you for your feedback. We would like to specifically address your concern regarding the procedure for inverting $ t \to \Lambda(t) $.
> > >
> > > We are very sorry if our previous explanation was unclear. Let us provide some additional clarifying details:
> > >
> > > First, $\Lambda$ is an increasing positive function, as the integrand is positive. Furthermore, according to Theorem 2.1, the integrand of $\Lambda$ is a continuous function, making $\Lambda$ a $C^1$ (continuously differentiable) increasing function. Since $\Lambda$ is monotonic increasing, it is injective, and thus an inverse exists. Moreover, this inverse function is also monotonic increasing and $C^1$.
> > >
> > > There are two options to obtain $ \Lambda^{-1} $. The first would be to obtain the set of points $ \\{(t_i, \Lambda(t_i))\\}\_{i=1}^N $ and fit a cubic interpolant to this set of points thus obtaining an approximation of $s=\Lambda(t)$. One would then have to perform some numerical strategy to invert this fitted function. The other option (this is the option that we perform in the paper), is to fit a cubic interpolant to the set of points $\\{ (\Lambda(t_i), t_i) \\}\_{i=1}^N$. Notice that we have swapped the order of $ t_i $ and $ \Lambda(t_i) $. When we fit a cubic interpolant to this set of points we directly obtain an approximation of $t= \Lambda^{-1}(s)$ __without ever having to perform any inversion__.
> > >
> > > We hope this additional detail clarifies our procedure and addresses your concern about the method of inverting $\Lambda$.
> > >
> > > Thank you again for your feedback, and we hope this explanation provides the clarity needed.

---

> > > > ### Comment · Area_Chair_jmmm · 2024-08-12
> > > > **Does reviewer 8ZVm have any further thought?**
> > > >
> > > > I thank both the reviewers and the authors for their contributions.
> > > > -AC

---

### Official Review · Reviewer_hc4h · 2024-07-07

**Soundness:** 3
**Presentation:** 3
**Contribution:** 3
**Rating:** 7
**Confidence:** 3

**Summary:**

A popular way to sample a target distribution is to run a "predictor-corrector" SDE, which additively combines two processes whose stationary distribution is the target: a Langevin (or "corrector") process and a reverse-diffusion (or "predictor") process.

How to discretize this SDE without deterioriating the approximation quality of its terminal distribution is the topic of this paper. The authors derive a discretization cost (Eq 13), which is the Fisher divergence between two distributions $\textemdash$ one obtained running the "predictor-corrector" process in continuous-time and the other obtained by running the "predictor" process only in discrete-time $\textemdash$ multiplied by the velocity of the "corrector" process.

This cost can be approximates using the estimated scores (section 3.2) and the authors optimize this cost with respect the schedule. They illustrate numerically the optimized schedules on image datasets and verify that they yield better approximations of the target distribution (measured in FID).

**Strengths:**

The paper is clearly written and derives a principled way to derive an optimal schedule for the popular "predictor-corrector" SDE, of which "annealed Langevin dynamics" and a "reverse diffusion process" are a special case. This is of interest to the sampling and diffusion models communities.

The appendix has nice, additional experiments assessing the optimal schedule and 1D and image datasets.

The authors are also clear about the limitations of their approach, namely that it assumes perfect estimation of the scores.

**Weaknesses:**

Including in the Appendix something like Figure 10 but for ImageNet, CIFAR, or CelebA would be interesting.

**Questions:**

It is unclear to me if in Eq 57, $J_t$ requires computing the Hessian of $\log p_t(\cdot)$: if so, wouldn't that be an expensive quantity to compute in order to obtain the optimal schedule?

**Limitations:**

Yes.

---

> ### Author Rebuttal · Authors · 2024-08-07
>
> We thank the reviewer for their comments. We agree that more visual comparisons of the different schedules would be illustrative. To this end, we will include standard diffusion progression plots, comparing the image diffusion path for CIFAR and CelebA on different schedules. In a potential camera-ready version, we would include one such plot in the main body of the text and all comparisons in the appendix. We greatly appreciate the reviewer’s comment, which has helped improve the presentation of our schedule optimisation.
>
> _It is unclear to me if in Eq 57, $J_t$ requires computing the Hessian of $\log p_t $: if so, wouldn't that be an expensive quantity to compute in order to obtain the optimal schedule?_
>
> We thank the reviewer for highlighting this important computational point. In our implementation, we have formulated an estimator (Proposition B.1) that completely avoids computing the full Hessian. This proposition is currently in the appendix, but we state it here for completeness:
>
> **Proposition B.1:** Let $ F_{t,t'} $ be the predictor map given by the forward Euler discretisation (8) of the probability flow ODE. For $ N \in \mathbb{N} $, let $ \hat{J}_{t,N}(x) $ be the Jacobian of the Hutchinson trace estimator (Hutchinson, 1989) for $ \nabla(\nabla \log p_t(x)) $ at $ x \in X $ and $ t \in [0, 1] $:
>
> $$
> \hat{J}_{t,N}(x) = \frac{1}{N} \sum_n \nabla(v_n^T J_t(x) v_n),
> $$
>
> If $ \Delta t $ is small enough such that:
>
> $$
> \Delta t \cdot \text{Tr} \left( f(t)I - 2g(t)\nabla \log p_t(x) \right) < 1,
> $$
>
> then, as $ N \to \infty $, the following limit exists almost surely:
>
> $$
> \nabla \log \det \nabla F_{t'}(x) = -\Delta t g(t)^2 \lim_{N \to \infty} \hat{J}_{t,N}(x) + O(\Delta t^2).
> $$
>
> Here, we only require the trace of the Hessian, so we need to compute only the diagonal entries, ensuring that the memory cost remains linear with respect to the parameters. Additionally, we have utilised a Hutchinson trace estimator for this term, which can be efficiently implemented in standard deep learning libraries.
>
> In our provided code, we have explicitly implemented this in PyTorch and have found that this computation is scalable to image datasets. This approach has enabled us to compute our predictor-optimised schedules, something infeasible if we had needed to compute the full Hessian. Our estimator performs well, producing schedules that achieve good FID scores. This was a subtle computational challenge that was non-trivial to overcome in our work.
>
> In a potential camera-ready version, we will clarify this approximation and emphasise that the computational resources required are significantly less than those needed for computing the full Hessian. We will also make clear in the main text that we have a scalable estimator for this term and do not require computing the Hessian matrix. We thank the reviewer for bringing this point to our attention and helping improve the quality of our paper.

---

> > ### Comment · Reviewer_hc4h · 2024-08-11
> > **Answer to authors**
> >
> > Thanks to the authors for their response.

---

> > > ### Author Response · Authors · 2024-08-11
> > >
> > > We thank the reviewer once more for their assessment of our work. Thank-you!

---

### Official Review · Reviewer_KRNd · 2024-07-13

**Soundness:** 3
**Presentation:** 3
**Contribution:** 3
**Rating:** 6
**Confidence:** 2

**Summary:**

The paper proposes an improved discretization schedules based on a novel cost measure that they proposes. The proposed method can update a given schedule to achieve greater sample quality as is demonstrated with solid experiments.

**Strengths:**

Strengths
* The proposed method is novel and can adapt to most given SGM model to update its discretization schedule to achieve better sample quality. The computation of the proposed method is also tractable.

**Weaknesses:**

Weaknesses
* While the paper provides heuristic theoretical justification to the design of the cost function, a better way to justify that the updated schedule provides better sample may be to directly measure the difference between the learned density (using a discretization schedule) and the true density $p_0$ and point out that updated schedule can reduce this difference.

**Questions:**

* It seems the proposed method do not accelerate the backward sampling process since the updated schedule has the same number of timesteps. I wonder if the author have examined the computational efficiency of their Algorithm 2? Since when it comes to SGM, the running time is one of the major concerns.

**Limitations:**

See weaknesses and questions above.

---

> ### Author Rebuttal · Authors · 2024-08-07
>
> We thank the reviewer for their review and suggested improvements. We agree that providing a simple error plot for cases where the true density is known would be informative. In our diffusion model, we do not have the ability to query the density directly; however, we do predict the score. In Figure 4 of our paper, we make this comparison for our bimodal example of the scores at t=0. It can be noted here that the linear schedule devolves into learning a function dissimilar to the expected shape of the score for a bimodal distribution. To further illustrate this point, we have created a plot that evolves over training, showing the error of the score at time t=0, corresponding to our data distribution. We observe that with our schedule optimisation, we achieve a lower error than with a fixed linear schedule over the course of training. Additionally, we have computed the likelihood of the generated samples in this case where the true density is known. Similarly, we observe that during training, our optimised schedule generates datasets with higher likelihood than the fixed schedule case. We thank the reviewer for their suggestion, which has improved the quality of our paper, and we will include these improvements in a potential camera-ready submission.
>
>
>
> _Q: It seems the proposed method do not accelerate the backward sampling process since the updated schedule has the same number of timesteps. I wonder if the author have examined the computational efficiency of their Algorithm 2? Since when it comes to SGM, the running time is one of the major concerns._
>
> We thank the reviewer for their question. We do observe in our simple 1D experiment in Figure 1 that an optimised schedule can achieve better results with fewer discretisation points, improving runtime. To better convey this potential use case, we have added follow-up experiments based on the reviewer's previous suggestion to see if we can see a gain in FID with fewer data points for image generation.
>
> Additionally, we have conducted a refinement test to assess the computational benefit of optimising the schedule. For this, we took the rho3 schedule for CIFAR10 and computed the FID score for 18, 20, 30, 50, and 100 discretisation points. In this range, at 18 discretisation points, the rho3 schedule achieves an FID of 5.47, while at 100 discretisation points, it achieves an FID of 2.05. In comparison, the optimised schedule achieves an FID of 1.99 at 18 discretisation points. We further observe the same trend in the sFID, another image performance metric.
>
> In this experiment, we can see that optimising the schedule can lead to improved FID and sFID performance with fewer discretisation points. Furthermore, we compared all schedules with only 10 discretisation points. In this case, the optimised schedule outperforms all others in terms of FID and sFID, demonstrating its superior performance in the few discretisation point regime. We thank the reviewer for their suggestion to perform this interesting computational experiment. In a potential camera-ready submission, we would include this study in the main manuscript.

---

> > ### Author Response · Authors · 2024-08-12
> >
> > We would like to thank the reviewer once again for their comments, which have led to improvements in our experiments, including a refined analysis that demonstrates a computational gain with our method in the few-point discretisation regime. We would like to confirm if we have addressed all of the reviewer's comments and questions. Thank you!

---

> > > ### Comment · Reviewer_KRNd · 2024-08-13
> > >
> > > Yes，my concerns have been addressed. Thank you for your informative supplementary experiments. I would keep my positive score.

---

### Author Rebuttal · Authors · 2024-08-07

We thank all the reviewers for taking the time to read our paper and for their kind comments on the strengths of the presentation and methodology. We also thank the reviewers for providing constructive feedback to improve the paper for the potential camera-ready version. We will address each point made by the reviewers individually. Below, we would like to summarize the additional experiments we conducted based on the reviewer feedback comparing (1) performance metrics and (2) schedule refinements. Please see the rebuttal document for the results.

(1) Performance metrics:
We have implemented the sFID metric in addition to FID and KLUB on the CIFAR dataset, which is included in our rebuttal document. We have found that the results for sFID correlate strongly with the results for FID in this case. In a potential camera-ready version, we would incorporate this metric across all the datasets presented. We appreciate the reviewer’s input, which has helped substantiate the validity of our method and improve the results presented in our paper.

(2) Schedule refinements:
Additionally, we have conducted a refinement test to assess the computational benefit of optimising the schedule. For this, we took the rho3 schedule for CIFAR10 and computed the FID score for 18, 20, 30, 50, and 100 discretisation points. In this range, at 18 discretisation points, the rho3 schedule achieves an FID of 5.47, while at 100 discretisation points, it achieves an FID of 2.05. In comparison, the optimised schedule achieves an FID of 1.99 at 18 discretisation points. We observe the same trend in the sFID, another image performance metric. Notably, the optimised schedule maintains stable performance even with 10 discretisation points compared to a suboptimal schedule.

We would further like to briefly comment on importance of the predictor/corrector framework used in our work and the reasoning for presenting our work within it. We further would like to address why we include analysis for both predictor and corrector optimised schedules in our work.

Our predictor/corrector framework is based on the seminal work of Yang Song et al. [1,2] (also see [2,3]). It is crucial to justify the final cost that we use to find optimal schedules. We agree that the final cost coincides with the standard Fisher Divergence; however, without the predictor/corrector formalism, it would be unclear whether this cost is justified and what relation controlling the Fisher Divergence would have in controlling error along the trajectory of a diffusion model. Our work provides this link by showing that the Fisher Divergence cost can be derived by minimising the work done by the diffusion model update steps under the predictor-corrector framework, thereby providing the relationship between the Fisher Divergence and diffusion model sampling. Furthermore, without understanding this link between predictor/corrector and the Fisher Divergence, it would be unclear what weighting should be applied to the Fisher Divergence when computing the cost for different time points along the trajectory. In Section 3.3, we find that the variance of the applied noise is the appropriate scaling, which is derived by noting that Langevin corrector steps should have steps on the order of the scale of the target distribution, which directly implies a meaningful scaling for the Fisher Divergence. Without the predictor/corrector framework, it would be unclear which scaling to use.

Corrector-optimised schedules are indeed more computationally efficient, requiring only one additional function evaluation per training step without gradient tracking compared to using no schedule optimisation. Despite this minimal overhead, they provide performance similar to the predictor-optimised schedule, which requires higher-order gradients. We explicitly mention this in our experimental section (Lines 311-312), where we recommend the use of the more cost-effective corrector-optimised schedule for image datasets. However, we believe it is important to include the analysis of the predictor-optimised schedule for two main reasons: firstly, it is a natural extension of the corrector schedule, and understanding its computational trade-offs could be valuable for the research community; secondly, while the corrector-optimised schedule performs similarly on image datasets, the predictor-optimised schedule may offer advantages in other application domains. We agree that the corrector-optimised schedule's benefits should be more prominently highlighted for its simplicity, lower cost, and competitive performance. We propose making this recommendation earlier in the paper, particularly in Section 2.

- [1] Generative Modeling by Estimating Gradients of the Data Distribution, Yang Song, Stefano Ermon
- [2] Score-Based Generative Modeling through Stochastic Differential Equations, Yang Song et al
- [3] Elucidating the Design Space of Diffusion-Based Generative Models, Term Karras et al
- [4] The probability flow ODE is provably fast, Sitan Chen et al

---

### Decision · Program_Chairs · 2024-09-25

**Decision:**

Accept (poster)

**Comment:**

This work proposes an approach to select a good discretization schedule for simulating the backward generation process of diffusion model. The selection was guided by an theoretically grounded approximation of discretization cost. Although there were some concerns about implementation cost and idealized assumptions such as perfect score, overall the merits overweight the limitations, and I'm delighted to recommend an acceptance. The authors are encouraged to account for the discussions in a revision.